# Thermodynamics of autonomous optical Bloch equations

Samyak Pratyush Prasad[1,2], Maria Maffei[3] and Patrice A. Camati[3], Cyril Elouard[5] and Alexia Auffèves[1,2]⋆

1 MajuLab, CNRS-UCA-SU-NUS-NTU International Joint Research Laboratory
2 Centre for Quantum Technologies, National University of Singapore, 117543 Singapore, Singapore
3 Dipartimento di Fisica, Università di Bari, I-70126 Bari, Italy
4 CNRS, Grenoble INP, Institut Néel, Université Grenoble Alpes, 38000 Grenoble, France
5 Université de Lorraine, CNRS, LPCT, F-54000 Nancy, France

⋆ alexia.auffeves@cnrs.fr ,

## Abstract

Optical Bloch Equations (OBEs) are canonical equations of quantum open systems, that describe the dynamics of a classically driven atom coupled to a thermal bath. Their thermodynamics is highly relevant to establish fundamental energetic bounds of key quantum processes. A consistent framework is available in the regime where drives and baths can be treated classically, i.e. remain insensitive to the coupling with the atom. This regime, however, is not adapted to explore minimal energy costs, nor to measure atom-induced energy variations inside drives and baths – a key ability to directly measure and optimize work and heat exchanges. This calls for a new framework where the atomic back-action on drives and baths would be accounted for. Here we build such a framework suitable to analyze the situation where the atom, the drive and the bath form a joint autonomous system, the drive and the bath being parts of the same electromagnetic field. Our approach captures atom-field correlations at fundamental timescales, as well as the atomic back-action on the field. This allows us to define work-like (heat-like) flows as energy flows stemming from effective unitary dynamics induced by one system on the other (non-unitary dynamics induced by correlations). Time-integrated work-like and heat-like flows are shown to be directly measurable in the field, as changes of energy locked in the mean field and fluctuations, respectively. Our approach differs from standard open analyses by identifying an additional unitary contribution in the atom's dynamics, the self-drive, and its energetic counterpart, the self-work, which yields a tighter expression of the second law. We quantitatively relate this tightening to the extra-knowledge about the field state as compared with usual treatments of the atom as an open system. Our autonomous framework deepens the current understanding of thermodynamics in the quantum regime and its potential for energy management at quantum scales. Its predictions can be probed in state-of-the-art quantum hardware, such as superconducting and photonic circuits.

# 1 Introduction

The question of how to measure work and heat flows in the quantum realm has represented one of the biggest challenges of quantum thermodynamics since its early days [1–7]. In classical thermodynamics, distinguishing the work from the heat received by a system relies on characterizing the processes giving rise to these energy flows. Thus, work and heat flows are usually reconstructed from the evolution of the system of interest, which requires to record its trajectory. While applying such strategy has provided pioneering measurements of heat and work flows in classical stochastic thermodynamics [8–12], it can hardly be applied in the quantum realm, because of measurement back-action. Namely, monitoring the trajectory of a quantum system impacts its subsequent dynamics [13] as well as its thermodynamic balance [14], calling for alternative strategies.

In that respect, it has been proposed to measure *the integrated work flow* (resp. heat flow) directly inside their sources, by identifying it to the energy change of the drive (resp. the bath) coupled to the system [15–19]. Here, it is in principle enough to perform measurements on the bath and the drive at the initial and at the final time of the protocol. This strategy becomes considerably simpler than tracking the system's trajectory, provided the drive and the bath are small enough systems so that measurements can be carried out and energy changes induced by the system take non-negligible values. Tremendous progresses in experimental capacities have already allowed to detect such small energy variations in the system's environment, whether drive or bath [20–22].

Nevertheless, describing the dynamics in presence of such finite-energy environments is beyond the scope of standard (weak coupling) quantum open system theory where drives and baths are treated classically - i.e. by neglecting any perturbation of their states upon interacting with the system [23]. Actually, a consistent dynamic and thermodynamic treatment rather requires to "close" the *formerly open* quantum systems. Namely, one should model the systems, drives and baths as the parties of a globally closed and isolated quantum system, taking into account their respective back-actions on one another. In such an autonomous picture, the parties should not have predetermined roles such as heat source, work sources, or working body. To make this obvious, it is instructive to consider the case of a quantum gate involving a qubit driven by a resonant pulse [24, 25]. Pulses of infinite energy qualify as ideal work sources, providing energy and no entropy to the qubit. Conversely, finite energy pulses get entangled with the qubit, changing its entropy along the process – while the qubit back-acts on the driving pulses, also acting as an energy and entropy source. This back-action gives rise to corrections to the drive which cannot be neglected anymore, such that the pulse hardly

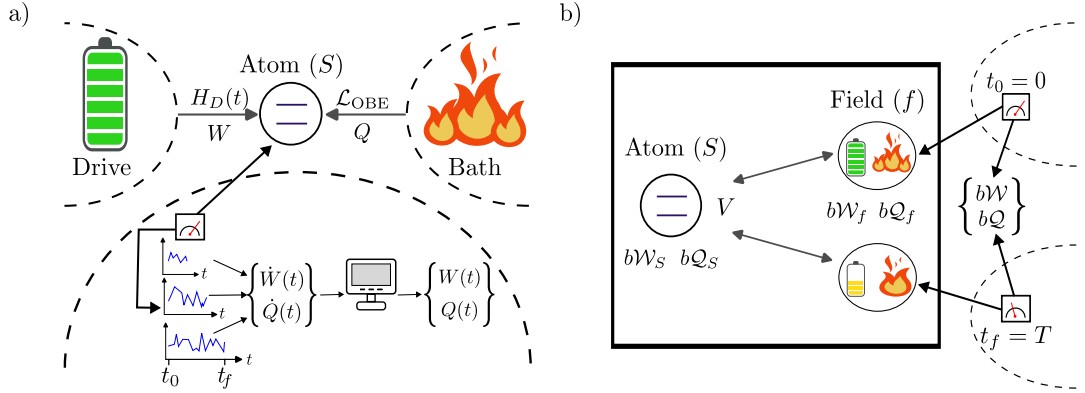

Figure 1: Schematics of a) the open and b) autonomous model, where the atom dynamics obeys the optical Bloch equations. a) The atom is coupled to a classical drive and to a thermal bath, receiving work $W$ and heat $Q$ from them respectively. Multiple tomographies of the atom are needed to measure the work and heat flows, which are then time-integrated between the initial time $t_0$ and the final time $t_f$. b) The atom, the drive and the bath form an autonomous system such that the total energy is conserved. Work-like ($b\mathcal{W}$) and heat-like ($b\mathcal{Q}$) quantities can be directly related to changes of the field amplitude and fluctuations between the initial and final time.

qualifies as an ideal work source.

This blurring of well-established thermodynamic roles calls for a new thermodynamic paradigm tailored for autonomous situations. Here the parties should play symmetric roles, while heat-like (work-like) flows should be defined with respect to the very nature of the processes inducing them. In this spirit, preliminary propositions have related heat (work) to energy exchanges induced by correlating (entropy-preserving) processes [26–31]. It is the purpose of this article to lay out the foundations of this paradigm shift. We shall consider a universal, textbook situation which can be both described as an autonomous, closed bipartite system, and as a driven open-quantum system weakly coupled to a Markovian bath.

With this aim, we focus on the canonical case of a coherently driven two-level atom weakly coupled to a thermal bath, whose dynamics is captured by the Optical Bloch Equations (OBEs) (See Fig. 1). The OBEs are interesting as they are hardware-universal, describing as well neutral atoms, superconducting qubits, quantum dots, or trapped ions [32–35]. They also provide a solid basis to explore a broad range of questions of fundamental and practical interest in quantum optics and quantum technologies, like quantum interfaces and quantum gates [36–38]. Therefore, their thermodynamics is highly relevant to explore and optimize the energy cost of quantum technologies [39].

Deriving the OBEs is textbook in the open-system paradigm [23,40], as they correspond to a famous example of the Lindblad equation. The thermodynamic analysis of the driven-dissipative atom has been conducted in various regimes [41–44]. In these analyses, the heat flow (the work flow) is defined as the power received from the bath (the classical drive). Minimal energy costs for atomic manipulations are identified with the heat irreversibly wasted in the bath. While thermodynamically consistent, these analyzes do not encompass any physical model for the drive, and do not keep track of the back-action of the system on the bath. Therefore, they cannot capture their actual energy and entropy changes, nor support consistent definitions of work-like and heat-like flows in the regime of finite energy for the drive and bath, let alone consistent protocols to measure them.

In this paper, we circumvent these limitations by focusing on an autonomous situation giving rise to the OBEs. A convenient physical scenery is provided by an atom coupled to driving pulses propagating in a circuit, as in circuit Quantum ElectroDynamics (QED) [34, 45–47] or in waveguide QED [48–50]. Remarkably, the electromagnetic field contained in the circuit encompasses both the drive and the bath. We build a framework to analyze the fully autonomous atom-field dynamics and thermodynamics, which unlocks the possibility of defining and measuring heat-like and work-like flows directly in the field. Additional information available in the field state and captured by our framework gives rise to a tighter expression of the second law, opening perspectives for energy savings with respect to the open description, both by enabling the use of low power pulses.

We first show that, in the regime of emergence of the OBEs, the autonomous dynamics of the joint atom-field system can be mapped onto a collision model [51, 52]. Importantly, this mapping allows us to keep track of the field evolution, as well as of the atom-field correlations, a major progress with respect to the state of the art and a key capacity to capture the back-action of each system on the other. Unlike when a non-autonomous collision model is pheno-menologically introduced to emulate the action of an environment [52–56], here the total Hamiltonian is time-independent, ensuring global energy conservation. However within a single collision, each system is driven by an effective time-dependent Hamiltonian entailing the reduced state of the other system, giving rise to a work-like energy exchange. The remnant term captures the effect of the correlations, giving rise to heat-like energy exchange. In this situation, the work-like flow (the heat-like flow) takes a remarkably simple form, as it equals the change of coherent (incoherent) power of the field induced by its coupling to the atom. These quantities are routinely accessible in -dyne and spectroscopic experiments, proving the operationality of the autonomous framework. As a matter of fact, these definitions have already been postulated in a handful of theoretical works [57–59], with the heuristic argument that the coherent part of the output field can be potentially reused to perform work on another emitter. They have also given rise to first direct measurements of heat-like and work-like flows in the autonomous scenery of circuit QED [60] and quantum optics with semiconducting quantum dots [31]. Our findings demonstrate they are rooted in a general and consistent framework, which can serve as a paradigm to characterize the nature of energy flows within closed autonomous systems.

In a second step, we quantitatively compare the open and the autonomous approaches. We show that they solely differ by an additional term in the effective drive on the atom – the self-drive – and its energetic counterpart, the self-work. The self-drive is induced by the coherent component of the radiation locally emitted by the atom, that our model allows to capture. We show that the energy carried by this local coherent component corresponds to the self-work. This quantity is usually not distinguished from the heat contribution in the open framework. This extra-component can be reused to drive other qubits, naturally reducing the amount of wasted heat. This yields a new expression of the second law that is tighter than the one obtained with open heat and work flows [44, 52]. We quantitatively relate the smaller entropy production to the increase of knowledge about the field state, that our framework accounts for. To make the comparison more concrete, we conduct it on two concrete situations routinely encountered in a quantum optics lab: a $\pi/2$ pulse, and a continuous driving.

Our results shed new light on fundamental mechanisms of quantum optics and quantum thermodynamics, and on key processes for light-based quantum technologies. Importantly, they provide the proper theoretical tools to analyze energy exchanges between quantum systems of interest, drives and baths, beyond the standard classical treatment of the latter. They can be probed on all experimental platforms modeled by waveguide or circuit QED, such as superconducting qubits [61] or semi-conducting quantum dots in micro-pillar cavities [31]. More generally, our autonomous collision model unlocks the possibility of exploring the impact

of correlations on the wide range of quantum open systems compliant with collision models.

The outline of the paper in the following. We first recall the open approach leading to state of the art thermodynamic analysis of the OBEs (Sec. 2). This approach is based on an open quantum system model involving a classical drive and bath, which remain unsensitive to the atom. We then introduce our autonomous dynamical model (Sec. 3). We define the classical regime for the drive, beyond which correlations between light and matter must be tracked. To do so, we upgrade our framework in Section 4 to keep track of these correlations, and propose the thermodynamical analysis fitting this description (Sec. 5). We then apply our thermodynamic framework the case of a pulse, and of a steady-state atomic driving (Sec. 6). This allows us to single out in concrete situations the potential of the autonomous approach for energy savings in the quantum regime. We respectively devote the last two sections (Sec. 7 and 8) to the measurement of work and heat flows through spectrally-resolved observables of the field, and to a comparison with non-autonomous collision models.

## 2  Open approach

The OBEs model the dynamics of a classically driven atom coupled to a thermal bath. Their dynamics in the interaction picture reads:

$$\dot{\rho}_S = -\frac{i}{\hbar}[H_D(t), \rho_S(t)] + \mathcal{L}_{\text{OBE}}[\rho_S(t)], \tag{1}$$

where $\rho_S$ is the state of the atom characterized by its transition frequency $\omega_0$ and excited (resp. ground) state denoted by $|e\rangle$ (resp. $|g\rangle$). The bare Hamiltonian of the atom reads $H_S = (\hbar\omega_0/2)\sigma_z$ with $\sigma_z = |e\rangle\langle e| - |g\rangle\langle g|$. $H_D(t)$ is the Hamiltonian induced by the classical drive of frequency $\omega_L$. It reads

$$H_D(t) = \frac{i\hbar\Omega}{2}\left(e^{i(\omega_0-\omega_L)t}\sigma_+ - e^{-i(\omega_0-\omega_L)t}\sigma_-\right), \tag{2}$$

where we have introduced the Rabi frequency $\Omega$, which quantifies the strength of the atom-field coupling. $\mathcal{L}_{\text{OBE}}$ is a Completely Positive Trace Preserving map (CPTP) which captures the effect of the thermal bath characterized by its inverse temperature $\beta$ (thermal population $\bar{n}_{\text{th}}$) and atomic spontaneous emission rate $\gamma$. It reads

$$\mathcal{L}_{\text{OBE}}[\rho_S] = \mathcal{L}(\sqrt{\gamma(\bar{n}_{\text{th}}+1)}\sigma_+, \rho_S)$$
$$+ \mathcal{L}(\sqrt{\gamma\bar{n}_{\text{th}}}\sigma_-, \rho_S), \tag{3}$$

where we have defined the Lindblad form

$$\mathcal{L}(O, \rho) \equiv O\rho O^\dagger - \frac{1}{2}\{\rho, O^\dagger O\}. \tag{4}$$

We now briefly recall the result of textbook thermodynamic analyses of the OBEs [44,62–64]. Introducing the atom internal energy

$$U(t) \equiv \text{Tr}_S\{\rho_S(t)(H_S + H_D(t))\}, \tag{5}$$

its time derivative $\dot{U}(t)$ splits into two components:

$$\dot{U}(t) = \dot{W}(t) + \dot{Q}(t), \tag{6}$$

where the work (heat) flow $\dot{W}(t)$ ($\dot{Q}(t)$) characterizes the power coherently exchanged with the classical drive (the power exchanged with the thermal bath). Within the approximations leading to the OBEs, they read:

$$\dot{W}(t) = \mathrm{Tr}_S \left\{ \rho_S(t) \dot{H}_D(t) \right\}, \tag{7}$$

$$\dot{Q}(t) = \mathrm{Tr}_S \left\{ (H_S + H_D(t)) \mathcal{L}_{\mathrm{OBE}}[\rho_S(t)] \right\}. \tag{8}$$

This choice of splitting brings out a consistent expression of the second law of thermodynamics [30, 44, 65]. Indeed, one recovers an equivalent of Clausius inequality by defining the quantity $\Sigma(t) = \Delta S(t) - \beta Q(t)$, where $\Delta S = S(\rho_S(t)) - S(\rho_S(0))$ is the change in the von Neumann entropy of the atom and $Q(t)$, the total heat received by the atom between the initial time $t_0 = 0$ and the time of interest $t$. Introducing the relative entropy $D(\rho||\sigma) = \mathrm{Tr}\{\rho(\log\rho - \log\sigma)\}$ between $\rho$ and $\sigma$, one can rewrite $\Sigma$ as

$$\Sigma(t) = D\left( \rho(t) || \rho_S(t) \otimes \rho_f^\beta \right) \geq 0. \tag{9}$$

We have introduced $\rho_f^\beta$ the thermal equilibrium state of the bath at inverse temperature $\beta$. Owing to the positivity of the relative entropy, $\Sigma(t)$ has been interpreted as a thermodynamic entropy production. As the environment is Markovian, its rate of change $\dot{\Sigma} \geq 0$ can be seen as a manifestation of the second law in the quantum regime [44, 65, 66]. This entropy production allows to estimate fundamental work costs associated to atomic manipulations, such as quantum gates. In particular, it provides access to the value of the non-recoverable work, i.e. the irreversibly wasted heat, over a thermodynamic cycle $W_{\mathrm{irr}} = \beta^{-1} \Sigma(\tau_{\mathrm{cycl}})$ where $\tau_{\mathrm{cycl}}$ is the duration of the cycle. Like in the classical realm, limiting the production of entropy is thus expected to have quantitative technological implications as it may impact the energetic bill of fundamental quantum processes.

While thermodynamically consistent, the open framework gives rise to challenging experimental protocols as depicted in Fig. 1a). Firstly, the drive is not given any physical description, and the variations of the bath and drive states are not tracked. This restricts the scope of measurements predicted by this approach to the sole atomic observables, as it appears from Eqs.(7) and (8). Moreover, the framework provides expressions for work and heat *flows*, which must be integrated over the duration of the whole experiment, i.e. between $t_0 = 0$ and $t = t_f$, to access the total work and heat received by the atom. However, these flows at time $t$ involve the atomic state $\rho_S(t)$, which must therefore be extracted by tomography at time $t$ for all $t$ between $t = 0$ and $t = t_f$. As measurement back-action alters the atom's state, hence its subsequent dynamics, the tomography of the atom requires to repeat the experiment from $t_0 = 0$ and $t$, for all $t$ between $t = 0$ and $t = t_f$.

This complex and heavy protocol strongly motivates to develop new strategies to measure heat and work, based on direct measurements inside the drive and the bath (See Fig. 1b). This in turn requires to develop autonomous models of light-matter interaction encompassing a quantum description of the bath and the drive, which is the purpose of the next section.

## 3 Autonomous model

In this Section we show that the coupled dynamics of the atom and the electromagnetic field of a circuit can be mapped onto an autonomous collision model, in the regime of emergence of the OBEs. As we shall see in this approach, the field plays the role of the drive and the bath.

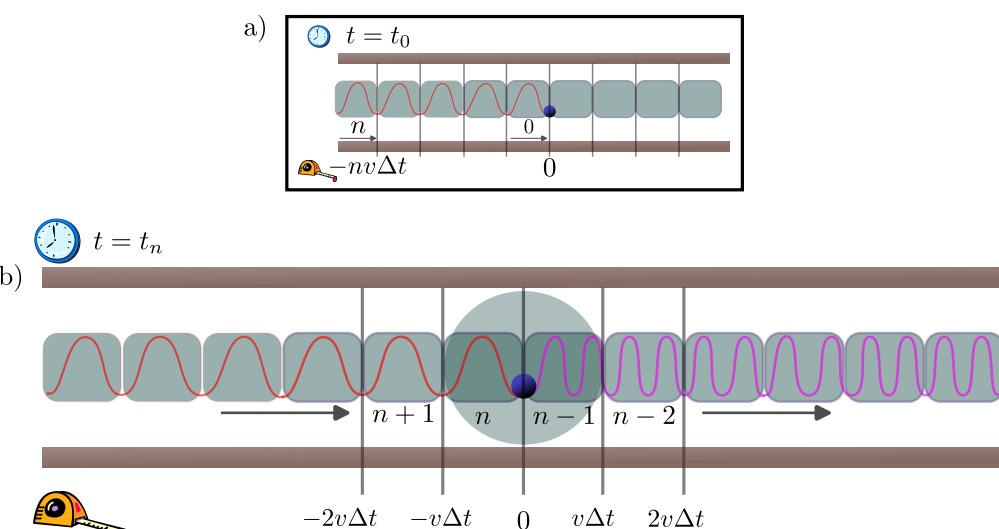

Figure 2: a) Joint atom-field system at the initial time $t = t_0$. The gray boxes represent the field collision units of size $v\Delta t$. The input units are located on $x \neq 0$, the $n^{\text{th}}$ unit being at position $x_n = -nv\Delta t$. In the case of Bloch equations, the input units are initially prepared in a displaced thermal field (See text). b) Evolution of the joint atom-field system between the times $t_n$ and $t_{n+1}$. The circular area denotes the interaction region. The atom starts to interact with the input unit ($n^{\text{th}}$ unit), and stops to interact with the output one ($(n-1)^{\text{th}}$ unit). Unlike the output units, the input units are prepared in a product of states that only differ by a relative phase.

## 3.1 Microscopic description

The situation we consider is depicted on Fig. 2. The atom stands at the position $x = 0$ of a one-dimensional (1D) reservoir of electromagnetic modes of momentum $k$, frequency $\omega_k$, and lowering operators $a_k$. All along the paper, we dub this collection of modes the 1D field, or more simply, the field. This situation is typical of circuit QED [34, 45–47] or waveguide QED [38, 50], and also captures the case of quantum emitters weakly coupled to adiabatically eliminated directional cavities [67]. All these situations lie at the basis of a wide range of light-matter functionalities (quantum interfaces, gates, networks, repeaters, etc) in a variety of physical platforms, from semiconducting quantum dots [31], superconducting circuits [20], to atomic physics [68], to name a few.

The autonomous atom-field dynamics is ruled by the time-independent Hamiltonian

$$H = H_S + H_f + V, \tag{10}$$

where $H_S$ is defined in Section 2 and the bare Hamiltonian of the field reads $H_f = \sum_k \hbar\omega_k a_k^\dagger a_k$. The field modes follow a linear dispersion relation around the atomic resonance, $\Delta\omega_k = v\Delta k$, with $\Delta\omega_k = \omega_k - \omega_0$ and $\Delta k = (k - k_0)$, $v$ being the group velocity and $\omega_0 = vk_0$. We shall use the rotating wave approximation, such that the coupling Hamiltonian $V$ reads

$$V = i\hbar \sum_{k\geq 0} g_k \left( a_k \sigma_+ - a_k^\dagger \sigma_- \right). \tag{11}$$

We have introduced $g_k$ as the coupling strength between the atom and the mode of frequency $\omega_k$, and the atomic operators $\sigma_- = |g\rangle\langle e|$, $\sigma_+ = \sigma_-^\dagger$. Following [29, 48, 67, 69–71], we only consider pulses propagating in one direction, restricting without loss of generality the study

to the modes of positive momentum $k \geq 0$. In this picture, pulses propagate from left to right. The atom-field dynamics is ruled by the following equation in the interaction picture with respect to $H_S + H_f$:

$$\frac{d\rho(t)}{dt} = -\frac{i}{\hbar}[V(t), \rho(t)]. \tag{12}$$

$\rho(t)$ $(V(t))$ is the joint quantum state of the atom-field system (the coupling Hamiltonian) in the interaction picture. Introducing the field operators $B(x,t) = \sum_k g_k e^{-i\Delta\omega_k(t-x/v)} a_k$, we rewrite the coupling Hamiltonian as

$$V(t) = i\hbar \left( B(0,t)\sigma_+ - \sigma_- B^\dagger(0,t) \right). \tag{13}$$

The operators $B(x,t)$ have the dimension of a frequency and are proportional to the local electric field at the position $x$. Finally, in the spirit of textbook microscopic derivations of the Lindblad equation of which the OBEs are a canonical example [23, 40], we shall take into account the finite spectral width of the coupling term characterized by the bandwidth $\Delta_{\text{bw}}$ over which the coefficients $g_k$ take non-negligible values. We introduce the atom spontaneous emission rate $\gamma = 2\pi \sum_k |g_k|^2 \delta_D(\Delta\omega_k)$, where $\delta_D$ stands for the Dirac distribution, and the correlation function $C(t,t') \equiv \sum_k |g_k|^2 e^{i\Delta\omega_k(t-t')}$. This function takes non-zero values over a time interval $|t - t'| < \tau_c$, where $\tau_c$ is the correlation time which scales like the inverse of the bandwidth $\Delta_{\text{bw}}$. Throughout the paper, we suppose the weak coupling hypothesis, which states

$$\gamma \tau_c \ll 1. \tag{14}$$

## 3.2 Coarse-grained collision model

Let us first write the atom-field evolution operator between $t_0 = 0$ and $t_f = T$ as a product of $N$ discrete-time dynamical maps of the form $\mathcal{U}_n = \mathcal{T} \exp\left\{ -i \int_{t_n}^{t_{n+1}} dt V(t)/\hbar \right\}$, where $t_n = \frac{nT}{N} = n\Delta t$ and $\mathcal{T}$ is the time ordering operator. We define the coarse-grained operators $b_n$

$$b_n = (\gamma\Delta t)^{-1/2} \int_{t_n}^{t_{n+1}} dt B(0,t), \tag{15}$$

which verify the commutation relation

$$[b_n, b_m^\dagger] = \frac{1}{\gamma\Delta t} \int_{t_n}^{t_{n+1}} dt \int_{t_m}^{t_{m+1}} dt' C(t,t'). \tag{16}$$

The weak coupling assumption [Eq.(14)] allows us to consider time intervals $\Delta t$ such that $\tau_c \ll \Delta t \ll \gamma^{-1}$. The inequality $\tau_c \ll \Delta t$ yields $[b_n, b_m^\dagger] = \delta_{m,n}$, ensuring that the $b_n$ correspond to independent modes (see Appendix A for the formal derivation of the discretization of $B(0,t)$ into $b_n$ modes). We then us the Magnus expansion [72, 73] to remove the time-ordering operator in $\mathcal{U}_n$ and write

$$\mathcal{U}_n = \exp\left\{ -\frac{i}{\hbar} \int_{t_n}^{t_{n+1}} du V(u) \right\} \equiv \exp\{-iV_n\}. \tag{17}$$

We have introduced the discrete evolution generator $V_n = i\sqrt{\gamma\Delta t}(b_n\sigma_+ - b_n^\dagger\sigma_-)$ (the technical details about the truncation of the Magnus expansion are provided in Appendix B).

From Eq.(17), we see that the atom-field evolution splits into a sequence of two-body interactions, or *collisions*, between the atom and collision units corresponding to the field modes $\{b_n\}$. The $n^{\text{th}}$ collision involves the operator $b_n$ and occurs during the time interval $[t_n, t_{n+1}]$. Below, we refer to the modes related to the operators $b_n$ as collision modes, or *collision units*. One can then exploit the inequality $\Delta t \ll \gamma^{-1}$ (naturally fulfilled in the usually considered case of a vanishing coarse-graining time, here enabled by the weak-coupling assumption) to expand $\mathcal{U}_n$ up to first order in $\gamma \Delta t$, i.e., at second order in $V_n$. Introducing $\rho_n \equiv \rho(t_n)$ and $\Delta_n \rho \equiv \rho_{n+1} - \rho_n$, we obtain the two first terms of the expansion (also known as the Dyson series),

$$
\begin{aligned}
\Delta_n \rho &= \Delta_n \rho^{(1)} + \Delta_n \rho^{(2)} + o(\gamma \Delta t), \\
\Delta_n \rho^{(1)} &= -i[V_n, \rho_n], \\
\Delta_n \rho^{(2)} &= -\frac{i}{2}[V_n, \Delta_n \rho^{(1)}] \\
&= \mathcal{L}(V_n, \rho_n).
\end{aligned}
\tag{18}
$$

Finally, we make the assumption that the atom and the collision units are initially prepared in a product state. As already noticed by [52, 74–76] for infinitely short collisions, the OBEs emerge for very specific initial unit states, i.e., displaced thermal states defined by $\eta_n = \mathcal{D}(\alpha_n)\rho_f^\beta \mathcal{D}^\dagger(\alpha_n)$, where $\mathcal{D}(\alpha_n) = \exp\{\alpha_n b_n^\dagger - h.c.\}$ displaces the $n^{\text{th}}$ collision unit by $\alpha_n$. The field thermal state of inverse temperature $\beta$ has been introduced above. It here verifies (See Appendix C) $\rho_f^\beta = \bigotimes_{n=0}^\infty \eta_n^\beta$, where $\eta_n^\beta \equiv e^{-\beta \hbar \omega_0 b_n^\dagger b_n}/(1 - e^{-\beta \hbar \omega_0})$ is the thermal state of the $n^{\text{th}}$ collision mode. As we show below, the initial displacement of the field gives rise to a coherent energy exchange with the atom, known as the classical Rabi oscillation, that corresponds to the open work. Conversely, the initial thermal component of the field yields a non-unitary evolution associated to an exchange of open heat.

**Input-Output Operators**

We now define operators related to the collision modes that are about to interact or have just interacted with the atom, i.e., input and output operators. To do so, we first relate the collision modes to the spatial position of the field. The linear dispersion relation in the waveguide allows us to write $B(x, t) = B(0, t - x/v)$, giving rise to another expression for the annihilation operators of the collision units, $b_n = \left(v\sqrt{\gamma \Delta t}\right)^{-1} \int_{x_{-(n+1)}}^{x_{-n}} dx B(x, 0)$ with $x_l = l v \Delta t$. Thus, the $n^{\text{th}}$ collision unit can equivalently be understood as the unit coupled to the atom between $t_n$ and $t_{n+1}$, or as the unit positioned between $x_{-(n+1)}$ and $x_{-n}$ at the initial time $t_0 = 0$ [see Fig. 2a)]. Writing the general expression of the operator acting on the mode positioned between $x_{n-1}$ and $x_n$ at any time $t_m$, $b_{x_n}(t_m) \equiv \left(v\sqrt{\gamma \Delta t}\right)^{-1} \int_{x_{n-1}}^{x_n} dx B(x, t_m)$, the field free dynamics is captured by the intuitive map $b_{x_n}(t_{m+1}) = b_{x_{n-1}}(t_m)$ [see Fig. 2b)], naturally yielding the output (input) operator:

$$
b_{\text{out(in)}}(t) \equiv \lim_{\gamma \Delta t \to 0} \frac{b_{x_1(x_0)}(t_n)}{\sqrt{\Delta t}}.
\tag{19}
$$

These operators written in the interaction picture should not be confused with the usual input and output operators defined in the Heisenberg picture [69]. Unlike the operators $b_n$ which are dimensionless, the operators $b_{\text{out(in)}}(t)$ have the dimension of $[t^{-1/2}]$.

### 3.3 Consistency with the OBEs and input-output theory

Before going further, we check that our model allows to recover the OBEs ruling the open dynamics of the atom as introduced in Section 2. To do so, we define

$$\Delta_n \rho^{(i)}_{S(f)} = \mathrm{Tr}_{f(S)}[\Delta_n \rho^{(i)}](i = 1, 2), \tag{20}$$

then write down the continuous limit of $\Delta_n \rho^{(i)}_{S(f)}/\Delta t$. On the atom side, this leads to the emergence of the OBEs [Eq. (1)], where the Hamiltonian $H_D(t)$ stems from the first term of the Dyson series. It features a constant external drive at the frequency $\omega_L$, and is unambiguously set by the initial state of the field [52] such that the the classical Rabi frequency $\Omega$ (see Eq. (2)) arises from the amplitude of the collision units as derived in Appendix C:

$$\frac{\Omega}{2} = \lim_{\gamma \Delta t \to 0} \sqrt{\frac{\gamma}{\Delta t}} \alpha_n e^{i(\omega_L - \omega_0)t_n}. \tag{21}$$

Conversely, the dissipator $\mathcal{L}_{\mathrm{OBE}}$ stems from the second order term of the Dyson series.

On the field side, one recovers the input-output equations which traditionally model the dynamics of an emitter coupled to the field of a circuit [69]. The demonstration goes as follows. Let us consider the field amplitude change between $t_n$ and $t_{n+1}$. During this time interval, only the $n^{\mathrm{th}}$ collision unit of the field interacts with the atom (See Section 3.2 and Fig. 2(b)). Hence, only this unit's amplitude changes and hence, the total field amplitude change between $t_n$ and $t_{n+1}$ is captured by the quantity $\mathrm{Tr}\{b_n \Delta_n \rho_f\}$. Eq. (18) yields the identity $\mathrm{Tr}\{b_n \Delta_n \rho_f\} = \langle b_{x_1}(t_{n+1})\rangle_{t_{n+1}} - \langle b_{x_0}(t_n)\rangle_{t_n}$ where we denoted $\langle O\rangle_t \equiv \mathrm{Tr}\{O\rho(t)\}$. Noticing that only the first order term contributes to the change of the field amplitude (see Appendix D for the detailed derivation), we recover a mean input-output equation in the continuous time limit:

$$\langle b_{\mathrm{out}}(t)\rangle_t = \langle b_{\mathrm{in}}(t)\rangle_t - \sqrt{\gamma}\langle\sigma_-\rangle_t. \tag{22}$$

### 3.4 Towards an operationally-relevant thermodynamic description

At this point we are ready to formulate the thermodynamic question we answer in this article. To do so, we apply the thermodynamic definitions established in the open framework, and show that they are only consistent in the classical regime for the drive and the bath, that our approach allows us to quantitatively define. We then present our strategy to extend them beyond the classical regime.

In this section and for the sake of pedagogy, we focus on the resonant case, which we define as a resonant drive $\omega_L = \omega_0$, or as the absence of a drive. We first express the open thermodynamic quantities defined in Section 2 in the context of input-output theory. Introducing the output (input) photon flow $\dot{N}_{\mathrm{out(in)}}(t) = \langle b^\dagger_{\mathrm{out(in)}}(t)b_{\mathrm{out(in)}}(t)\rangle_t$ allows us to rewrite Eqs.(6), (7) and (8) as

$$\dot{U}(t) = \hbar\omega_0(\dot{N}_{\mathrm{in}}(t) - \dot{N}_{\mathrm{out}}(t)) \tag{23}$$

$$\dot{W}(t) = -\hbar\omega_0 \dot{N}_{\mathrm{drive}}(t) \tag{24}$$

$$\dot{Q}(t) = -\hbar\omega_0 \dot{N}_{\mathrm{bath}}(t), \tag{25}$$

where $\dot{N}_{\mathrm{drive}}$ and $\dot{N}_{\mathrm{bath}}$ have the following expressions:

$$\dot{N}_{\mathrm{drive}} = -\sqrt{\gamma}\left(\langle\sigma_+\rangle_t\langle b_{\mathrm{in}}(t)\rangle_t + \langle\sigma_-\rangle_t\langle b^\dagger_{\mathrm{in}}(t)\rangle_t\right), \tag{26}$$

$$\dot{N}_{\mathrm{bath}} = \gamma\left(\left(\bar{n}_{\mathrm{th}} + \frac{1}{2}\right)\left(2\langle\sigma_+\sigma_-\rangle_t - 1\right) + \frac{1}{2}\right). \tag{27}$$

The meaning of the equations above is transparent. The open work flow $\dot{W}$ [Eq.(24)] is proportional to the photon flow $\dot{N}_{\text{drive}}$. It stems from the unitary driving process induced by the input field, that is known as the classical Rabi oscillation. This is a coherent energy exchange, as it appears from Eq.(26) which has the form of an interference term between the amplitude of the atomic radiation and that of the input field. Conversely the open heat flow $\dot{Q}$ [Eq.(25)] is proportional to the photon flow $\dot{N}_{\text{bath}}$ exchanged with the thermal component of the input field, which plays the role of a bath. In the case of a zero-temperature field, this distinction captures the splitting between stimulated emissions and absorptions on the one hand, and spontaneous emissions on the other.

We now define the classical limit of the drive as $\gamma|\langle b_{\text{in}}\rangle|^2 \gg 1$, i.e. $\dot{N}_{\text{drive}} \gg \dot{N}_{\text{bath}}$. Here the atom dynamics is purely unitary and ruled by the Hamiltonian $H_S + H_D(t)$. The change of the field energy simply equals the integrated work flow received by the atom as computed from Eq.(24). This identity allowed for pioneering direct measurements of work flows in circuit QED [20].

The purpose of this paper is to explore the thermodynamics of the atom-field interaction in the case of a finite input power, for which both $\dot{N}_{\text{drive}}$ and $\dot{N}_{\text{bath}}$ contribute to the atom-field power exchange. As it appears from their expressions [Eqs.(26) and (27)], these components cannot be distinguished based on the sole output field observables. The amount of open work (heat) exchanged between 0 and $t$ must be accessed via a series of atomic tomographies at each time step, as explained in Section II, depicted in Fig. 1 and argued in [60]. Beyond the practical obstacles they give rise to, the definitions stemming from the open approach are incompatible with a finite input power. Indeed, one expects the driving mode to get correlated with the atom, giving rise to entropy inputs which do not correspond to the behavior of an ideal work source. In addition, the impact of the interaction with the atom on the field's state cannot be neglected anymore. Corrections to the field-induced driving are therefore expected.

These limitations have lead some of us to propose an alternative approach to identify heat and work in this regime [57], which has been used in a handful of theoretical [59, 77] and experimental works [31, 60]. Work (heat) is defined as the change of energy carried by the coherent (incoherent) component of the field. Here the rationale is that the field induces a unitary drive of the atom, and therefore exerts some work on it, via its coherent component. This boils down to splitting the term $\langle \sigma_+ \sigma_- \rangle_t$ appearing in Eq.(27) into its mean field and fluctuations, $|\langle \sigma_- \rangle_t|^2 + \langle \delta\sigma_+ \delta\sigma_- \rangle_t$, and to let the mean field term contribute to the work flow. This definition matches the open work (heat) definition in the classical regime of high input power, where the coherent driving dominates. Moreover, it is operational as coherent (incoherent) field components can be distinguished in -dyne experiments, hence the direct measurements of these quantities in circuit QED [60] and quantum photonics [31].

In the two next sections, we show that these definitions are actually rooted in a broader, thermodynamically consistent framework. Unlike the standard framework of quantum thermodynamics which focuses on open systems, the new framework we build is tailored for a fully autonomous situation like the model proposed above. Our thermodynamic definitions are inspired from former attempts developed for composite autonomous systems [26, 27, 78]. In these approaches, the key idea is to define work-like (heat-like) flows as entropy-preserving (correlating) processes. As we show below, applying such a strategy confirms the heuristic definitions proposed and exploited in [31, 57, 59, 60, 77].

Building this framework requires in the first place to develop the capacity to track atom-field correlations inside our autonomous model: this is the purpose of the next section.

## 4  Tracking correlations

Here we refine the autonomous model introduced in Section 3 to keep track of the correlations forming during one collision. We first introduce this analysis in the general case, then we focus on the OBEs, where we unveil the impact of the correlations on the atom and on the field dynamics.

### 4.1  General case

Let us resume from the Dyson expansion [Eqs. (18)] and rewrite the first order term as

$$
\begin{aligned}
\Delta_n \rho^{(1)} = {}& \Delta_n \rho_S^{(1)} \otimes \rho_f(t_n) \\
& + \rho_S(t_n) \otimes \Delta_n \rho_f^{(1)} + \Delta_n \chi^{(1)}.
\end{aligned}
\tag{28}
$$

The first order terms $\Delta_n \rho_{S(f)}^{(1)}$ have been introduced in Eqs.(20). They allow us to express the change of the atom-field correlations during the $n^{\text{th}}$ collision:

$$
\Delta_n \chi^{(1)} \;=\; -i[V_n - \langle V_n \rangle_S - \langle V_n \rangle_f, \rho_n].
\tag{29}
$$

Injecting Eq. (28) into Eqs.(18) splits the second term of the Dyson series into three new terms:

$$
\begin{aligned}
\Delta_n \rho^{(2,S)} &= -\frac{1}{2}[V_n, [\langle V_n \rangle_f, \rho_n]], \\
\Delta_n \rho^{(2,f)} &= -\frac{1}{2}[V_n, [\langle V_n \rangle_S, \rho_n]], \\
\Delta_n \rho^{(2,\chi)} &= -\frac{1}{2}[V_n, [V_n - \langle V_n \rangle_f - \langle V_n \rangle_S, \rho_n]].
\end{aligned}
\tag{30}
$$

Eqs.(30) unveil three new mechanisms in the evolution of each subsystem, which are hidden in the standard approach. The term $\Delta_n \rho_S^{(2,S)} = \text{Tr}_f[\Delta_n \rho^{(2,S)}]$ (resp. $\Delta_n \rho_f^{(2,f)} = \text{Tr}_S[\Delta_n \rho^{(2,f)}]$) can be rewritten as $\Delta_n \rho_S^{(2,S)} = -\frac{1}{2}[\langle V_n \rangle_f, [\langle V_n \rangle_f, \rho_S(t_n)]]$ (resp. $\Delta_n \rho_f^{(2,f)} = -\frac{1}{2}[\langle V_n \rangle_S, [\langle V_n \rangle_S, \rho_f(t_n)]]$). Thus, these expressions capture the second-order terms in the expansion of the unitary evolution generated by the effective Hamiltonian $\langle V_n \rangle_{f(S)}$. Conversely, crossed terms like $\Delta_n \rho_S^{(2,f)} = \text{Tr}_f[\Delta_n \rho^{(2,f)}]$ (resp. $\Delta_n \rho_f^{(2,S)} = \text{Tr}_S[\Delta_n \rho^{(2,S)}]$) can be rewritten as $\Delta_n \rho_S^{(2,f)} = -\frac{i}{2}\text{Tr}_f\{ [V_n, \Delta_n \rho_f^{(1)} \otimes \rho_S(t_n)]\}$ (resp. $\Delta_n \rho_f^{(2,S)} = -\frac{i}{2}\text{Tr}_S\{ [V_n, \Delta_n \rho_S^{(1)} \otimes \rho_f(t_n)]\}$ ). Each term features an effective drive of $S(f)$, mediated by the first-order evolution of $f(S)$: we dub such a mechanism a *self-drive*. Both the second order drive and the self-drive correspond to Hamiltonian processes in the reduced dynamics of the atom and the field. Conversely, terms of the form $\Delta_n \rho_{S(f)}^{(2,\chi)}$ capture the remnant non-unitary evolution of the atom (field) state induced by the atom-field correlations.

Finally, this ability to track correlations invites us to rewrite the evolution of the atom-field system along one collision, by isolating the contributions of the correlations/unitary evolutions up to the second order in the Dyson expansion:

$$
\Delta_n \rho^{(1)} + \Delta_n \rho^{(2)} = \Delta_n \rho^{\chi} + \Delta_n \rho^{\otimes}.
\tag{31}
$$

We have introduced $\Delta_n \rho^{\chi}$ and $\Delta_n \rho^{\otimes}$:

$$
\begin{aligned}
\Delta_n \rho^{\otimes} &= \Delta_n \rho_S^{(1)} \otimes \rho_f(t_n) + \rho_S(t_n) \otimes \Delta_n \rho_f^{(1)} \\
&\quad + \Delta_n \rho^{(2,S)} + \Delta_n \rho^{(2,f)}, \\
\Delta_n \rho^{\chi} &= \Delta_n \chi^{(1)} + \Delta_n \rho^{(2,\chi)}.
\end{aligned}
\tag{32}
$$

The analysis above is valid for any bipartite system $S - f$ complying with a Magnus expansion treatment. Now focusing on the regime of emergence of the OBEs, we exploit the splitting captured by Eq. (31) to explore how the correlations impact the atom's (field's) evolution.

## 4.2  Imprint on the atom

To highlight the role of correlation in the atom's dynamics, we re-derive the master equation of the atom (see Appendix E):

$$\dot{\rho}_S(t_n) = \lim_{\gamma \Delta t \to 0} \mathrm{Tr}_f \left\{ \frac{\Delta_n \rho^{\chi} + \Delta_n \rho^{\otimes}}{\Delta t} \right\}$$
$$= -\frac{i}{\hbar} \left[ H_D(t) + \mathcal{H}_S^s(t), \rho_S(t) \right] + \mathcal{L}_{t,S,\chi}. \tag{33}$$

The uncorrelated part of the dynamics captured by $\Delta_n \rho^{\otimes}$ yields two unitary contributions. The only non-zero first order contribution associated with $\Delta_n \rho_S^{(1)}$ is unchanged with respect to the Dyson series [Eq. (20)], and yields the classical driving term $H_D(t)$ already present in the standard form of the OBEs [Eq.(1)]. In addition, our approach singles out an additional unitary contribution, stemming from the second-order term $\Delta_n \rho_S^{(2,f)}$, that we call self-driving Hamiltonian, or *self-drive*:

$$\mathcal{H}_S^s(t) = -i\hbar \frac{\gamma}{2} (\langle \sigma_- \rangle_t \sigma_+ - \langle \sigma_+ \rangle_t \sigma_-). \tag{34}$$

The self-drive is induced by the coherent component of the radiation locally emitted by the atom, which scales like the atom coherence $\langle \sigma_- \rangle_t$. It can take finite values in the absence of any external drive, provided the atom initially carries coherences in the energy basis. At zero temperature, it yields a rotation of the atom's state in the Bloch sphere during the transient regime of spontaneous emission. Reciprocally, the correlations are responsible for a non-unitary term $\mathcal{L}_{t,S,\chi}$ stemming from $\Delta_n \rho^{(2,\chi)}$, which can be cast as:

$$\mathcal{L}_{t,S,\chi} = \mathcal{L}\left( \sqrt{\gamma(\bar{n}_{\mathrm{th}} + 1)} (\sigma_+ - \langle \sigma_+ \rangle_t), \rho_S \right)$$
$$+ \mathcal{L}\left( \sqrt{\gamma \bar{n}_{\mathrm{th}}} (\sigma_- - \langle \sigma_- \rangle_t), \rho_S \right), \tag{35}$$

This term constitutes a new Lindbladian involving jump operators shifted by their instantaneous average value.

We stress that Eq. (33) yields the exact same state evolution as the OBEs [Eq. (1)]. This property can be retrieved by noting that these two Lindblad equations are connected by one of the transformations of the Lindblad operators and Hamiltonian leaving the dynamics invariant [23]. Another insight is provided by realizing that the dissipator $\mathcal{L}_{\mathrm{OBE}}$ [Eq. (1)] stems from the second order term $\Delta \rho_n^{(2)}$ of the Dyson expansion, which splits into a unitary and a non-unitary part to give rise to the maps captured by the self-drive $\mathcal{H}_S^s(t)$ and the new Lindbladian $\mathcal{L}_{t,S,\chi}$, respectively.

Pioneering studies in quantum open systems conducted in Heisenberg picture have singled out the *self-reaction* as the physical process giving rise to the self-drive [79] – namely, the result *on the atom* of the polarisation of the field by the atom. Because it treats symmetrically the atom and the field, and provides full access to the field state, our framework sheds new light on the self-reaction, and draws so far overlooked consequences of this mechanism in the weak atom-field coupling regime. As we show below, the imprint of the self-drive on the field takes the form of a tiny amount of coherent energy, the self-work. These observable thermodynamic consequences (see Section 5) constitute a key difference between the open and the autonomous approach.

### 4.3 Imprint on the field

Let us now focus on the impact of the correlations on the field's evolution. The mean input-output equation [Eq.(22)] ruling the variation of the field's amplitude remains unaltered by the new splitting since it solely involves the first order term $\Delta_n \rho_f^{(1)}$. This is not the case for the change of the photon flow, which involves the second order term $\Delta_n \rho_f^{(2)}$. Eq.(23) can be rewritten as:

$$\dot{N}_{\text{out}} - \dot{N}_{\text{in}} = \dot{N}^\otimes + \dot{N}^\chi. \tag{36}$$

We have introduced the change of the photon flow stemming from Hamiltonian (correlating) processes $\dot{N}^\otimes$ ($\dot{N}^\chi$), which verify (see Appendix D for the derivation):

$$\dot{N}^\otimes \equiv \lim_{\gamma \Delta t \to 0} \frac{\text{Tr}[b_n^\dagger b_n \Delta_n \rho^\otimes]}{\Delta t}$$
$$= |\langle b_{\text{out}}(t) \rangle_t|^2 - |\langle b_{\text{in}}(t) \rangle_t|^2, \tag{37}$$

$$\dot{N}^\chi \equiv \lim_{\gamma \Delta t \to 0} \frac{\text{Tr}[b_n^\dagger b_n \Delta_n \rho^\chi]}{\Delta t}$$
$$= \langle \delta b_{\text{out}}^\dagger(t) \delta b_{\text{out}}(t) \rangle_t - \langle \delta b_{\text{in}}^\dagger(t) \delta b_{\text{in}}(t) \rangle_t. \tag{38}$$

The term $\delta b_{\text{out(in)}}(t) = b_{\text{out(in)}}(t) - \langle b_{\text{out(in)}}(t) \rangle_t$ stands for the fluctuations of the output (input) field. Eqs. (36), (37) and (38) unveil an intimate relation between mean field evolution and Hamiltonian processes on the one hand, and between fluctuations and correlating processes on the other.

We now consider the physical meaning of the flows $\dot{N}^\otimes$ and $\dot{N}^\chi$. As it appears on Eq.(37), $\dot{N}^\otimes$ captures the change of the mean photon flow contained in the coherent component of the field. For this reason, we shall refer to it as the *coherent photon flow*. From the mean value of the output field [Eq.(22)], we get

$$\dot{N}^\otimes = \sqrt{\gamma} \left( \langle \sigma_+ \rangle_t \langle b_{\text{in}}(t) \rangle_t + \langle \sigma_- \rangle_t \langle b_{\text{in}}^\dagger(t) \rangle_t \right)$$
$$+ \gamma \left| \langle \sigma_- \rangle_t \right|^2 \tag{39}$$

The first contribution to $\dot{N}^\otimes$ has already been identified as $\dot{N}_{\text{drive}}$ [Eq.(26)] and captures the exchange of photons with the coherent component of the input field through classical Rabi oscillations. Moreover, $\dot{N}^\otimes$ features an additional term scaling like $|\langle \sigma_- \rangle_t|^2$. This contribution is the imprint of the atom's self-drive, which is manifested by an additional coherent radiation emitted by the atom in the bath. As we show below, it carries the main thermodynamic difference between the open and the autonomous approach. Conversely, the flow $\dot{N}^\chi$ induced by the atom-field correlating processes [Eq.(38)] reads

$$\dot{N}^\chi = \gamma \langle \delta \sigma_+(t) \delta \sigma_-(t) \rangle_t, \tag{40}$$

where $\delta \sigma_-(t) = \sigma_-(t) - \langle \sigma_- \rangle_t$ stands for the fluctuations of the atomic dipole and $\sigma_-(t) = e^{-i \omega_0 t} \sigma_-$. This corresponds to a purely incoherent atomic radiation in the bath – we shall refer to it as the *incoherent photon flow*. The change of the field mean value and fluctuations can be measured through -dyne or photon counting experiments. The splitting we propose between Hamiltonian and correlating processes thus corresponds to physical, experimentally accessible quantities. As we show below, these quantities exactly map the splitting between work-like and heat-like flows.

## 5  Thermodynamics of the autonomous scenario

We now draw the consequences of the dynamical framework built above for the thermodynamics of the autonomous atom-field system. To make a clear distinction with the standard, open approach, we shall use curly symbols to denote all energetic quantities defined within the autonomous framework.

### 5.1  General considerations

Firstly, we define the atom and field internal energies as $\mathcal{U}_j(t) \equiv \mathrm{Tr}_j\left\{H_j\rho(t)\right\}$, where $j \in \{S, f\}$, which ensures a symmetric treatment of the two parties. The coupling energy term is defined as $\mathcal{V}(t) = \mathrm{Tr}\{V(t)\rho(t)\}$. Recalling that the total Hamiltonian $H_{\mathrm{tot}} = H_S + H_f + V$ is time-independent, the total energy $\mathcal{U}_{\mathrm{tot}} = \mathcal{U}_S(t) + \mathcal{U}_f(t) + \mathcal{V}(t)$ is conserved, thus equal to the initial amount of energy injected in the joint atom-field system. Such global energy conservation allows us to analyze individual energy changes as resulting from energy exchanges between the atom, the field, and the coupling term.

Building on the analysis of a collision performed above, we now consider the energy changes locally experienced by the atom and the field during the $n^{\mathrm{th}}$ collision. In full generality, they split into two different contributions, giving rise to two symmetrical expressions, equivalent to a first law for each system:

$$\Delta_n\mathcal{U}_j = \Delta_n[b\mathcal{W}_j] + \Delta_n[b\mathcal{Q}_j], \tag{41}$$

for $j \in \{S, f\}$, with

$$\Delta_n[b\mathcal{W}_j] = \mathrm{Tr}\left\{H_j\Delta_n\rho^{\otimes}\right\} \tag{42}$$

and

$$\Delta_n[b\mathcal{Q}_j] = \mathrm{Tr}\left\{H_j\Delta_n\rho^{\chi}\right\}. \tag{43}$$

The amount of energy $\Delta_n[b\mathcal{W}]_j$ received by the system $j$ during the $n^{\mathrm{th}}$ collision stems from the effective unitary exerted by the other system. It is entropy-preserving, and reminiscent of a work flow. To make a clear distinction with the concept of work established for open systems (corresponding to Eq.(7) in the case of the OBE), we dub the corresponding energy flow in the continuous time limit $b\dot{\mathcal{W}}_j(t) \equiv \lim_{\gamma\Delta t\to 0}\Delta_n[b\mathcal{W}_j]/\Delta t$ a "bipartite work flow" or more simply $b$-work flow. In the classical limit of the drive, where negligible correlations build up between the two parties during their interaction, the dynamics of system $j$ is unitary, and it only receives $b$-work: in this limit, the $b$-work matches the standard work in quantum thermodynamics, i.e. the change of energy of a system evolving under a time-dependent Hamiltonian [63]. In the case of the OBEs, this classical limit corresponds to a high-power drive as explained in Section 3.4. Then, the only energy flow received by the atom is the open work flow [Eq.(7)]. Conversely, $b\dot{\mathcal{Q}}_j(t) \equiv \lim_{\gamma\Delta t\to 0}\Delta_n[b\mathcal{Q}_j]/\Delta t$ is induced by the correlations between the two parties during one collision – we refer to it as the "bipartite heat flow" or $b$-heat flow. The considerations above can be applied to any bipartite dynamics captured by an autonomous description as described in Section 3. The rest of this section is devoted to the thermodynamic analysis of the autonomous OBEs.

### 5.2  First laws

As in Section 3.4, here we consider the resonant regime defined by $\omega_L = \omega_0$, or by the absence of a drive. To keep the text light, the expressions in the non-resonant regime are provided in

Appendix F, as well as all technical calculations. In this resonant regime, the time-derivative of the coupling term $\dot{\mathcal{V}}(t)$ vanishes, yielding the simple expression for the energy conservation:

$$\dot{\mathcal{U}}_f(t) = -\dot{\mathcal{U}}_S(t) = \hbar\omega_0(\dot{N}_{\text{out}}(t) - \dot{N}_{\text{in}}(t)). \tag{44}$$

Moreover, the resonant condition imposes that the $b$-work ($b$-heat) flows received by the atom and the field are opposite. From the general considerations above, it should not come as a surprise that the splitting between $b$-work and $b$-heat flows reflects the splitting between mean values and fluctuations captured by Eqs.(37) and (38). Indeed we show in Appendix F that the $b$-work flows are proportional to the change of the coherent photon flow $\dot{N}^{\otimes}$ [Eq. (37)]:

$$b\dot{\mathcal{W}}_f(t) = -b\dot{\mathcal{W}}_S(t) = \hbar\omega_0\dot{N}^{\otimes} \tag{45}$$
$$= \hbar\omega_0\left(|\langle b_{\text{out}}(t)\rangle|^2 - |\langle b_{\text{in}}(t)\rangle|^2\right)$$

Conversely, the $b$-heat flows are proportional to the incoherent photon flow [Eq. (38)], and read

$$b\dot{\mathcal{Q}}_f(t) = -b\dot{\mathcal{Q}}_S(t) = \hbar\omega_0\dot{N}^{\chi} \tag{46}$$
$$= \hbar\omega_0(\langle\delta b_{\text{in}}^{\dagger}(t)\delta b_{\text{in}}(t)\rangle - \langle\delta b_{\text{out}}^{\dagger}(t)\delta b_{\text{out}}(t)\rangle)$$

These remarkably simple and intuitive relations unlock the possibility of measuring directly the *integrated* amounts of $b$-work and $b$-heat exchanged between the atom and the field. This is because unlike for the atom, measuring observables of the output field at time $t$ does not impact its subsequent evolution. Thus, the $b$-work is simply equal to the change of energy stored in the field's coherent component, or *field's coherent energy*. In the same way, the $b$-heat is equal to change of energy stored in the field's fluctuations, or *incoherent energy*. These quantities are routinely extracted from -dyne measurements. In this picture, the field appears as a bookkeeper of the processes by which it has exchanged energy with the atom. In particular, these findings shed new light on the phase-space deformations of the field, which appear as direct consequences of correlating processes – even if these correlations no longer exist. In this view, the $b$-heat has been interpreted as a witness of past correlations in [29, 31].

To conclude and as already mentioned in the introduction, these definitions were theoretically postulated in [29, 57–59, 77], with the heuristic argument that the coherent part of the output field can be potentially reused to perform work on another emitter. They have given rise to first direct measurements of heat and work flows in the circuit QED platform [60] and in the semi-conducting platform [31]. The autonomous thermodynamic framework proposed here provides a general consistency to this body of work, and extends its validity to the case of arbitrary driving fields and finite bath temperatures.

## 5.3 Self-work

We now compare the open and the autonomous approach from a thermodynamic viewpoint, first focusing on the energetic differences. We recall that the open work flow received by the atom is proportional to the photon flow $\dot{N}_{\text{drive}}$ coherently exchanged with the driving mode [Eq.(24)]. Conversely, the $b$-work flow is proportional to the total coherent photon flow $\dot{N}^{\otimes}$. Comparing Eqs.(24), (39), and (45) yields

$$\dot{W} = b\dot{\mathcal{W}}_S - b\dot{\mathcal{W}}_S^s, \tag{47}$$

where we have introduced the self-work flow

$$b\dot{\mathcal{W}}_S^s = -\gamma\hbar\omega_0\left|\langle\sigma_-\rangle_t\right|^2. \tag{48}$$

The self-work is equal to the work exerted by the self-drive [Eq. (34)] on the atom. From Eq.(48), the self-work is always negative in the physical situation under study. It corresponds to the coherent contribution of the energy radiated by the atom in the bath. Thus, it is considered as a contribution to heat in the open approach. This is captured by the equations comparing $b$-heat flows and open heat flow,

$$
\begin{aligned}
\dot{Q}(t) &= b\dot{\mathcal{W}}_S^s(t) - b\dot{\mathcal{Q}}_f(t) \\
&= b\dot{\mathcal{W}}_S^s(t) + b\dot{\mathcal{Q}}_S(t)
\end{aligned}
\tag{49}
$$

Unlike the open work, the self-work flow can take finite values, even in the absence of a coherent component in the input field. The presence of self-work was predicted in [57] and experimentally reported in [31] in the case where the field is initially in the vacuum state. In the presence of a coherent component in the input field, measuring the self-work is more tricky and has not been conducted yet. It can be extracted in principle, by measuring independently the $b$-work and the open work flow - with the drawback that measuring the open work flow is a heavy process as already argued in Section II, depicted in Fig. 1 and reported in [60]. Another strategy can be used if the atom weakly radiates in other leaky modes than the 1D channel. Denoting as $\eta$ the ratio of coupling between the leaky modes and the 1D channel, and measuring the coherent power $P_{leak}$ radiated in the leaky modes yields the expression of the self-work $b\dot{\mathcal{W}}_S^s = \eta P_{leak}$.

Eqs. (47) and (49) convey that the self-work can be interpreted as a reusable energy, which is unlocked by the increased knowledge over the field provided by the one-dimensional environment. We now relate these intuitions to the second law of thermodynamics, and explore how potential energy savings impact entropy production in the autonomous framework.

## 5.4 A tighter second law

Let us first recall the expression of the open entropy production between the initial time $t_0 = 0$ and $t$, $\Sigma(t) = D\left(\rho(t)||\rho_S(t) \otimes \rho_f^{\beta}\right)$ [Eq. (9)]. It can be decomposed as

$$
\Sigma(t) = I(t) + D\left(\rho_f(t)||\rho_f^{\beta}\right),
\tag{50}
$$

where $I(t) = S(\rho_S(t)) + S(\rho_f(t)) - S(\rho(t))$ is the mutual information between the atom and the field at time $t$. The splitting in Eq. (50) suggests an information-theoretic interpretation of the second law, in which irreversibility is attributed to the lack of information about the joint atom-field state at time $t$, as quantified by the right-hand side terms [65, 66]. The term $I(t)$ accounts for the ignorance of the atom-field correlations, consistent with the fact that we solely access local quantities. Conversely, the relative entropy $D\left(\rho_f(t)||\rho_f^{\beta}\right)$ measures the informational distance between the reduced output field state at time $t$ and the estimate of its state. In the open approach, this estimate simply corresponds to the thermal equilibrium state $\rho_f^{\beta}$, assumed to be unperturbed by the interaction with the atom to derive the weak-coupling master equations.

In contrast, keeping track of the field dynamics induced by the coupling to the atom conveys more information about the reduced field state at time $t$. This additional information can be used to modify the field reference state appearing in Eq. (50), letting the $b$-heat appear. Namely, we now take the new reference state to be the field state propagated by the effective unitary induced by the atom. This effective unitary corresponds to a sequence of displacements on each collision unit, between the initial time $t_0 = 0$ and the time of interest $t$ (see Appendix C and H for the derivation):

|  | Open Approach | Closed Approach |
|---|---|---|
| Atom | $\dot{\rho}_S = -\frac{i}{\hbar}[H_D(t), \rho_S(t)] + \mathcal{L}_{\text{OBE}}[\rho_S(t)]$ | $\dot{\rho}_S = -\frac{i}{\hbar}\left[H_D(t) + \mathcal{H}_S^s(t), \rho_S(t)\right] + \mathcal{L}_{t,S,\chi}$ <br> $\mathcal{L}_{t,S,\chi} = \mathcal{L}_{\text{OBE}}[\rho_S(t)] - \left(-\frac{i}{\hbar}\left[\mathcal{H}_S^s(t), \rho_S(t)\right]\right)$ |
| Field | $\dot{N}_{\text{out}} - \dot{N}_{\text{in}} = \dot{N}_{\text{drive}} + \dot{N}_{\text{bath}}$ | $\dot{N}_{\text{out}} - \dot{N}_{\text{in}} = \dot{N}^{\otimes} + \dot{N}^{\chi}$ <br> $\dot{N}^{\otimes} = \dot{N}_{\text{drive}} + \gamma|\langle\sigma_-\rangle_t|^2$ <br> $\dot{N}^{\chi} = \dot{N}_{\text{bath}} - \gamma|\langle\sigma_-\rangle_t|^2$ |
| Work and Heat received by the Atom | $\dot{W} = -\hbar\omega_0\dot{N}_{\text{drive}}$ <br> $\dot{Q} = -\hbar\omega_0\dot{N}_{\text{bath}}$ | $b\dot{\mathcal{W}}_S = -\hbar\omega_0\dot{N}^{\otimes} = \dot{W} + b\dot{\mathcal{W}}_S^s$ <br> $b\dot{\mathcal{Q}}_S = -\hbar\omega_0\dot{N}^{\chi} = \dot{Q} - b\dot{\mathcal{W}}_S^s$ <br> $b\dot{\mathcal{W}}_S^s = -\gamma\hbar\omega_0|\langle\sigma_-\rangle_t|^2$ |
| Measurement Protocol | $\dot{W}$ and $\dot{Q}$ not directly measurable in the field <br><br> Inferred via "stroboscopic" tomography of the atom (See Fig.1a) | $b\dot{\mathcal{W}}_S = \hbar\omega_0\left(|\langle b_{\text{in}}(t)\rangle|^2 - |\langle b_{\text{out}}(t)\rangle|^2\right)$ <br> $b\dot{\mathcal{Q}}_S = \hbar\omega_0(\langle\delta b_{\text{in}}^{\dagger}(t)\delta b_{\text{in}}(t)\rangle)$ <br> $-\hbar\omega_0(\langle\delta b_{\text{out}}^{\dagger}(t)\delta b_{\text{out}}(t)\rangle)$ <br><br> Directly measurable in the field via single-dyne measurement of the field (see Section 5) |
| Entropy Production | $\Sigma = I(t) + D(\rho_f\|\rho_f^{\beta})$ <br><br> $\implies \Delta S_S - \beta Q(t) \geq 0$: Clausius Inequality | $b\Sigma = I(t) + D(\rho_f\|\mathcal{D}(t)\rho_f^{\beta}\mathcal{D}^{\dagger}(t))$ <br><br> $\implies \Delta S_S - \beta b\mathcal{Q}_S(t) \geq 0$: Tighter expression of Second Law |

Table 1: Table summarizing the differences between the open and the autonomous frameworks in the resonant regime, from both the dynamical and thermodynamical viewpoint.

$$\mathcal{D}(t) = \prod_{n=0}^{N}\mathcal{D}(\varphi_n), \tag{51}$$

where we have introduced $\mathcal{D}(\varphi_n) = \exp\{(\varphi_n b_n^{\dagger} - \varphi_n^* b_n)\}$ and $\varphi_n = -\sqrt{\gamma\Delta t}\langle\sigma_-\rangle_{t_n}$. The new field reference state is then $\mathcal{D}(t)\rho_f^{\beta}\mathcal{D}^{\dagger}(t) = \bigotimes_{n=0}^{N}\mathcal{D}(\varphi_n)\eta_n^{\beta}\mathcal{D}^{\dagger}(\varphi_n)$. The modified entropy production reads

$$b\Sigma(t) = D\left(\rho(t)\middle\|\rho_S(t)\bigotimes_{n=0}^{N}\mathcal{D}(\varphi_n)\eta_n^{\beta}\mathcal{D}^{\dagger}(\varphi_n)\right)$$

$$= I(t) + D\left(\rho_f(t)\middle\|\bigotimes_{n=0}^{N}\mathcal{D}(\varphi_n)\eta_n^{\beta}\mathcal{D}^{\dagger}(\varphi_n)\right). \tag{52}$$

Crucially, the rate of change of $b\Sigma(t)$ gives rise to an inequality featuring the $b$-heat flow received by the field (see Appendix H for the derivation):

$$b\dot{\Sigma}(t) = \dot{S}_S + \beta\, b\dot{\mathcal{Q}}_f \geq 0, \qquad\qquad (53)$$

which we interpret as the formulation of the second law consistent with the autonomous approach. At resonance, we retrieve a Clausius-like inequality, where $b\Sigma = \Delta S_S - \beta\, b\mathcal{Q}_S \geq 0$. In this case, using Eq. (49), it is straightforward to notice that $-\dot{Q}(t) \geq -b\dot{\mathcal{Q}}_S(t)$ as $b\dot{\mathcal{W}}_S^s \leq 0$ and hence, $b\dot{\Sigma} \leq \dot{\Sigma}$ - leading to a smaller entropy production rate and also a tighter expression of the second law.

This smaller entropy production is directly related to a more accurate accounting of the non-equilibrium resources present in the output field state, to include the work exchanges via displacements of the temporal modes. As they contain a coherent component, they can be used in principle to coherently drive other emitters, providing work to them [31,59,77]. They are therefore relevant if one attempts to recycle the field after its interaction with the atom. When even more knowledge (or control) about the local state of the bath can be assumed, a similar approach was used by some of us to derive an expression of the second law in general autonomous quantum setups, accounting for all non-equilibrium resources [30]. Proposing concrete protocols to recycle this energy, however, is beyond the scope of this paper.

The main elements of the comparison between the autonomous and the open approach are provided in Table 1. It shows that we have fulfilled our initial motivations to provide a thermodynamically consistent, operational and potentially useful framework capturing the dynamics and the thermodynamics of a driven atom coupled to a thermal bath, beyond the classical regime for the drive (the bath). To make the comparison more concrete, we devote the next Section to the study of a few physical situations.

# 6  Applications

In this section, we consider two situations captured by the OBEs, concerning respectively the transient dynamics (a $\pi/2$ pulse) and the steady-state regime of the atom.

## 6.1  Thermodynamics of a $\pi/2$ pulse

We first study the thermodynamic balance in the case of a $\pi/2$ pulse. The atom is initially prepared in its ground state $|g\rangle$. At time $t_0 = 0$, it is coupled to the 1D field. The driving pulse has a square shape, its Rabi frequency $\Omega$ and duration $\tau$ are tuned such that $\Omega\tau = \pi/2$, and the field is at zero temperature. Thus the ideal transformation reached in the limit $\Omega/\gamma \gg 1$ is $|g\rangle \rightarrow |+\rangle$. After interacting with the pulse, the atom spontaneously emits a (partially) coherent superposition of 0 and 1 photon in the field.

Fig. 3 displays the open and autonomous thermodynamic flows as a function of time, and the corresponding integrated quantities for $\Omega/\gamma = 10$ and $\Omega/\gamma = 2$. The energetic advantage of using the autonomous framework is obvious. While the pulse is applied, the open work cost overcomes the $b$-work, while the amount of $b$-heat wasted in the bath is lower than the open heat. In both cases, the self-work is responsible for the reduction of the energetic bill. In the second step, the atom spontaneously releases its energy. There is no extraction of open work associated to this relaxation as there is no external drive. However, the initial coherence present in the atom gives rise to the extraction of a finite self-work in the field, which again, lowers the amount of wasted heat.

In Fig. 4, we have plotted the thermodynamic quantities integrated over the duration of the pulse, the relaxation, and both, as a function of the ratio $\Omega/\gamma$. Increasing this ratio brings the

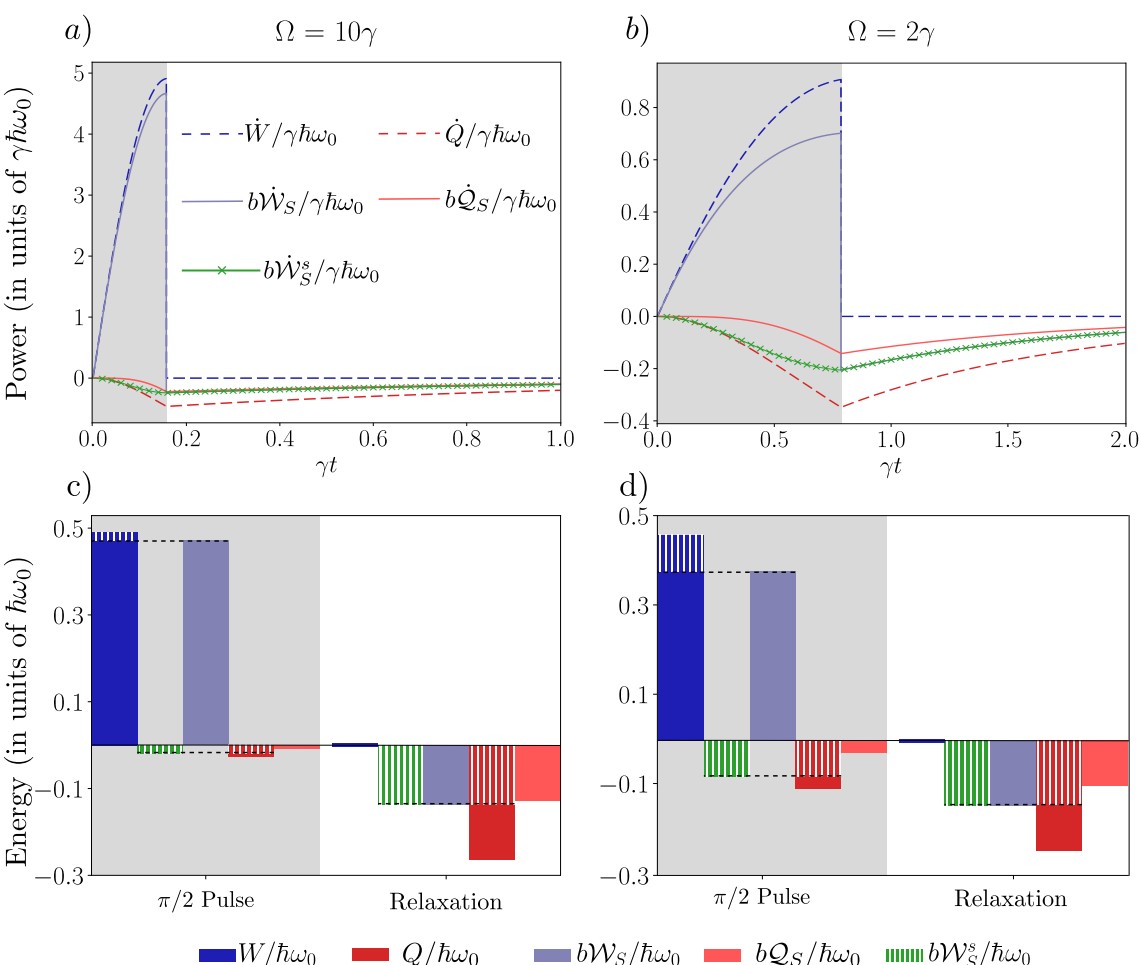

Figure 3: Thermodynamic flows in the open and in the autonomous approach (in units of $\gamma\hbar\omega_0$) in the case of a $\pi/2$ pulse (See text). The atom is resonantly excited by a square coherent pulse of strength $\Omega$ for a duration $\tau = \pi/2\Omega$ (grey region), then left to relax through spontaneous emission process (white region). The dashed blue (red) line denotes the open work (heat). The purple (red) solid line denotes the flow of $b$-work ($b$-heat) exchanged with the drive. The crossed green lines denote the self-work flow. a) Strong drive $\Omega = 10\gamma$ b) Weak drive $\Omega = 2\gamma$. Insets: Bar plots of the total open work and heat, $b$-work and $b$-heat and self-work received by the atom during the interaction with the $\pi/2$ pulse and relaxation stages. The striped area denotes the proportion of self-work with respect to the different energetic quantities.

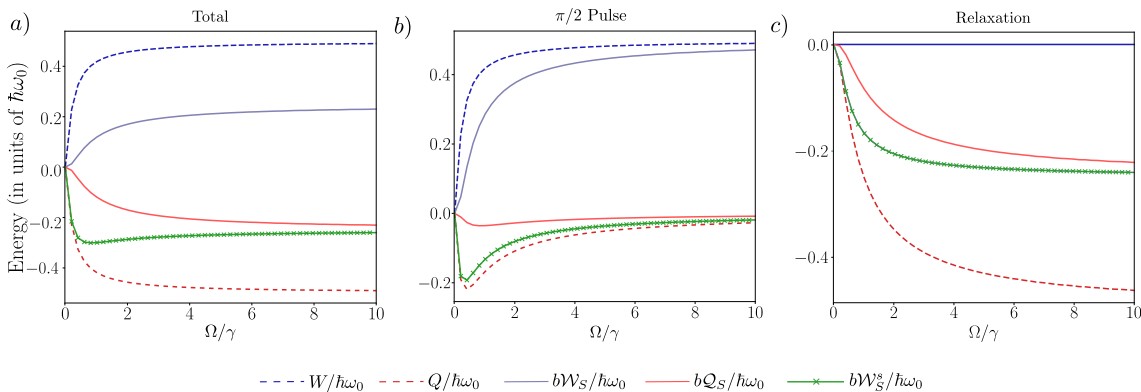

Figure 4: Time-integrated thermodynamic flows against the ratio $\Omega/\gamma$ in the case of a $\pi/2$ pulse (See text). The dashed blue and red lines denote the open work and heat, while the solid purple and red lines denote the $b$-work and $b$-heat received by the atom. The crossed-green line denotes the self-work. The integration is performed over a) the total duration of the protocol (pulse+relaxation) b) the $\pi/2$ pulse c) the relaxation.

pulse closer to ideality, which corresponds to a vanishing amount of wasted open heat during the driving step. For this reason, the self-work vanishes, as well as the difference between open and autonomous quantities. Conversely, the maximal self-work is reached for $\Omega \sim \gamma$. During the relaxation step, no open work is received by the atom, but self-work is provided to the field. Here the amount of self-work increases with the ratio $\Omega/\gamma$, as it allows to prepare the purest coherent atomic superposition $|+\rangle$. In this case, the self-work lowers the amount of wasted open heat by half a photon.

The considerations above can easily be generalized to the case of a Hadamard gate, by taking an arbitrary atom's initial state. Depending on whether the driving pulse provides or receives work (i.e., depending on the initial atom's population), the self-work lowers the work cost paid by the field or increases the work received by the field. This gives a glimpse of the potential of the autonomous framework for energy savings in fundamental quantum processes.

## 6.2 Steady-state thermodynamics

Let us now suppose that the steady-state regime of the OBEs is reached. Namely, we consider a continuous, resonant driving and focus on the open and the autonomous thermodynamic quantities reached after a time $t_\infty \gg \gamma^{-1}$, where the mean values of the observables of the atom and the output field become independent of time. Steady-state driving is prevalent in quantum optics in the context of reservoir engineering [80], quantum engines [81], or more recently, for the generation of non-Gaussian resources using optical non-linearities [82]. A convenient parameter to analyze this steady-state situation is the saturation parameter $s$ [40], which has the following expression in the resonant regime

$$s = \frac{2\Omega^2}{\gamma^2 \left(2\bar{n}_{\text{th}} + 1\right)^2}. \tag{54}$$

Whether in the autonomous or in the open approach, the work and the heat flows received by the atom are equal and opposite. We chose to plot in Fig. 5a) the steady-state work flows for both approaches, and in Fig. 5b) their difference, i.e. the self-work, as a function of $s$ and for three different temperatures. As it appears on Fig. 5a), the open work flow does solely depend on the temperature through the saturation parameter, and steadily increases as a function of

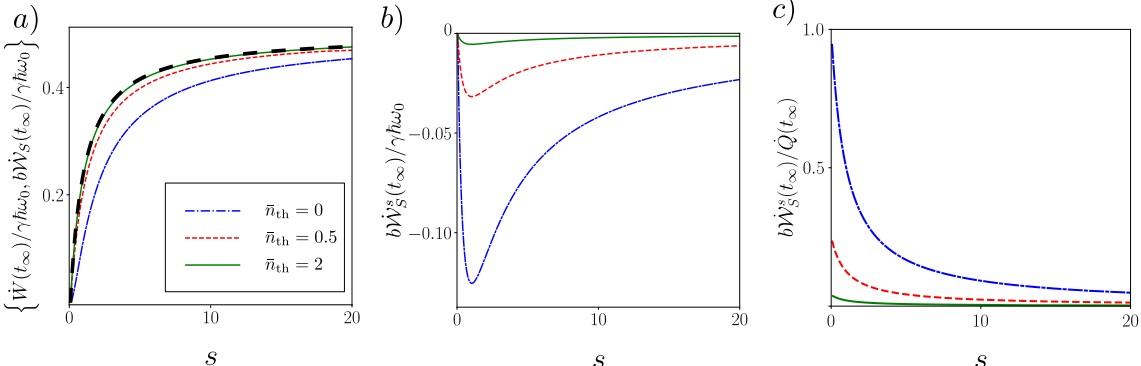

Figure 5: Steady state plots illustrating Eq. (47) with energetic quantities in units of $\gamma\hbar\omega_0$. The $b$-work and self-work rates are depicted by colored lines, dash-dotted blue line for $\bar{n}_{\text{th}} = 0$, dashed red line for $\bar{n}_{\text{th}} = 0.5$, and solid green line $\bar{n}_{\text{th}} = 2$. a) Resonant $b$-work flow and work flow (black dashed line) are plotted against the saturation parameter $s$. The $b$-work converges towards the work (which does not depend explicitly on the temperature) for increase in temperature and $s$. b) Resonant self-work flow is plotted against $s$. It depicts the difference between the $b$-work and the work. c) coherent ratio $r$ at steady state against $s$.

$s$. For high saturation parameters, the steady-state atom's coherence $\langle\sigma_-\rangle_\infty$ vanishes and the open work flow reaches a typical rate of half a photon per atomic lifetime. Conversely, both the $b$-work flow and the self-work flow explicitly depend on temperature. This is expected as the self-work flow scales like the steady-state atom's coherence given by

$$|\langle\sigma_-\rangle_\infty|^2 = \frac{b\dot{\mathcal{W}}_S^s}{-\gamma\hbar\omega_0} = \frac{1}{2(2\bar{n}_{\text{th}}+1)^2}\left(\frac{s}{(1+s)^2}\right). \tag{55}$$

Maximal values of the self-work flow are reached for a vanishing temperature, and for $s \sim 1$, i.e. $\Omega \sim \gamma$. Finally, remembering that in the steady-state $|\dot{Q}|_\infty \geq |b\dot{\mathcal{W}}_S^s|_\infty$, it is useful to define a coherent ratio as $r = \frac{|b\dot{\mathcal{W}}_S^s|_\infty}{|\dot{Q}|_\infty}$. This ratio characterizes the coherence in the steady-state power emitted in the bath which can, in principle, be used to provide $b$-work to another qubit. Hence, this ratio could also quantify the potential of recycling the field emitted in the bath. We have plotted in Fig. 5 the coherent ratio, which reaches 1 for $s \to 0$ and zero temperature. This regime has been dubbed the Heitler or Rayleigh regime, and captures the emission of a quasi-coherent field by an atomic dipole in the low excitation regime [83]. Conversely, $r$ decreases with $s$ and with temperature as both parameters lower the steady-state coherence of the atomic dipole.

# 7 Spectral analysis

In the autonomous thermodynamic framework, the splitting between $b$-work and $b$-heat is reflected in the splitting between coherent and incoherent photon flows, which can be traced back through time-resolved -dyne measurements of the field. However, this splitting can also be captured through the field spectral observables. It is the purpose of this section to explore this alternative path. Most of the calculations are presented in Appendix I. As above, we consider the resonant case where $\omega_0 = \omega_L$ and we focus on the steady-state regime of the OBEs, that is reached after driving the atom from $t_0 = 0$ to typical times $t = t_\infty \gg \gamma^{-1}$. We

first define the input (output) spectral density of flow as

$$\dot{S}_{\text{in(out)}}(\omega) = \frac{1}{\pi}\text{Re}\int_0^{t_\infty} d\tau e^{-i\Delta\omega\tau}G_{\text{in(out)}}(\tau), \tag{56}$$

where $\Delta\omega = \omega - \omega_0$, Re denotes the real part and we have introduced the steady-state input (output) field correlation functions

$$G_{\text{in(out)}}(\tau) = \left\langle b_{\text{in(out)}}^\dagger(t_\infty + \tau)b_{\text{in(out)}}(t_\infty)\right\rangle_{t_\infty}. \tag{57}$$

The spectral densities of flow can be measured through standard spectroscopic experiments performed on the input (output) fields. We trivially recover the identity $\int d\omega \dot{S}_{\text{in(out)}}(\omega) = \dot{N}_{\text{in(out)}}(t_\infty)$. We show in Appendix I that the correlation functions $G_{\text{in(out)}}(\tau)$ split to reveal the Hamiltonian (correlating) processes yielding them:

$$G_{\text{in(out)}}(\tau) = G_{\text{in(out)}}^\otimes(\tau) + G_{\text{in(out)}}^\chi(\tau), \tag{58}$$

$$G_{\text{in(out)}}^\otimes(\tau) = \left\langle b_{\text{in(out)}}^\dagger(t_\infty + \tau)\right\rangle_{t_\infty+\tau}\left\langle b_{\text{in(out)}}(t_\infty)\right\rangle_{t_\infty},$$

$$G_{\text{in(out)}}^\chi(\tau) = \left\langle \delta b_{\text{in(out)}}^\dagger(t_\infty + \tau)\delta b_{\text{in(out)}}(t_\infty)\right\rangle_{t_\infty}.$$

This naturally leads to a splitting in the difference of spectral densities of flow $\dot{S}_{\text{out}}(\omega) - \dot{S}_{\text{in}}(\omega) = \dot{S}^\otimes(\omega) + \dot{S}^\chi(\omega)$:

$$\dot{S}^\otimes(\omega) = \frac{1}{\pi}\text{Re}\int_0^{t_\infty} d\tau e^{-i\Delta\omega\tau}(G_{\text{out}}^\otimes(\tau) - G_{\text{in}}^\otimes(\tau)) \tag{59}$$

$$\dot{S}^\chi(\omega) = \frac{1}{\pi}\text{Re}\int_0^{t_\infty} d\tau e^{-i\Delta\omega\tau}(G_{\text{out}}^\chi(\tau) - G_{\text{in}}^\chi(\tau)), \tag{60}$$

where we check that $\int d\omega \dot{S}^{\otimes(\chi)}(\omega) = \dot{N}^{\otimes(\chi)}(t_\infty)$. For this reason we dub $\dot{S}^\otimes(\omega)$ ($\dot{S}^\chi(\omega)$) a coherent (an incoherent) spectral density of flow. Expressing the output field observable as a function of the atomic observables, we find that the coherent spectral density of flow is composed of a single frequency, which is the driving frequency (here equal to the atomic frequency):

$$\dot{S}^\otimes(\omega) = -\delta_D(\omega - \omega_0)\bigg[\frac{\Omega}{2}\left(\langle\sigma_-\rangle_{t_\infty} + \text{c.c.}\right)$$

$$-\gamma|\langle\sigma_-\rangle_{t_\infty}|^2\bigg]. \tag{61}$$

The presence of two terms in the coherent spectral density of flow is consistent with the structure of $\dot{N}^\otimes(t_\infty)$ [Eq.(39)]. The term scaling like $\Omega$ results from the coherent driving process. Conversely, the term scaling like $\gamma$ corresponds to the elastic component of the atomic emission spectrum, known at zero temperature as the Mollow triplet [40]. In Appendix I.2, we show that the inelastic component of the Mollow triplet is equal to the incoherent spectral density of flow $\dot{S}^\chi(\omega)$ when the temperature is set to zero. Generalizing to non-zero temperatures, we recover in Appendix I that $\dot{S}^\chi(\omega)$ has a three-peak structure centered around $\omega_0$ and of typical width $\gamma(1 + \bar{n}_{\text{th}})$.

Owing to their different spectral characteristics (Dirac-delta vs triplet of peaks of finite width), the coherent and the incoherent spectral densities of flow can be simply distinguished experimentally. We now exploit these results to provide a new way of measuring work and heat flows. To do so, we define the field energy spectral density as

$u_f(\omega, t) \equiv \sum_k \delta_D(\omega_k - \omega)\hbar\omega_k \mathrm{Tr}[a_k^\dagger a_k \rho(t)]$. As shown in Appendix I, the power spectral density verifies:

$$\dot{u}_f(\omega) = \hbar\omega\left(\dot{S}_{\mathrm{out}}(\omega) - \dot{S}_{\mathrm{in}}(\omega)\right), \tag{62}$$

which can be rewritten as

$$\dot{u}_f(\omega) = b\dot{w}_f(\omega) + b\dot{q}_f(\omega), \tag{63}$$

$$b\dot{w}_f(\omega) \equiv \hbar\omega\dot{S}^{\otimes}(\omega), \tag{64}$$

$$b\dot{q}_f(\omega) \equiv \hbar\omega\dot{S}^{\chi}(\omega). \tag{65}$$

We easily check that $\int d\omega\, b\dot{w}_f(\omega) = b\dot{\mathcal{W}}_f$ and similarly for the $b$-heat. Since the coherent and incoherent spectral densities of flow are experimentally accessible, we conclude that we can extract the values of the steady-state $b$-work and $b$-heat flows from a spectroscopic analysis of the input and output fields. These results provide another evidence of the operational value of our framework and enrich the set of experimental tools to characterize energy exchanges between light and matter.

# 8 Connection to the non-autonomous collision model

In this final section, we connect the energy balance of our autonomous model to the one derived for more standard non-autonomous collision models (also known as repeated interaction schemes) [52,53,84]. We extend the study to non-resonant cases $\delta = \omega_L - \omega_0 \neq 0$, where the coupling energy $\mathcal{V}(t)$ plays a non-trivial role in the energy balance:

$$\dot{\mathcal{U}}_S(t) + \dot{\mathcal{U}}_f(t) + \dot{\mathcal{V}}(t) = 0. \tag{66}$$

The definitions of $\mathcal{U}_{S(f)}(t)$ and $\mathcal{V}(t)$ are given in Section 5.1, yielding $\dot{\mathcal{V}} = -\frac{i}{\hbar}\mathrm{Tr}\left\{V(t)[H_f + H_S, \rho(t)]\right\}$. In the energy balance Eq. (66), the coupling term hence effectively behaves as an independent party which can store and provide energy. We show in Appendix F that the system, field and coupling energy flows take in our model the simple expressions:

$$\dot{\mathcal{U}}_S(t) = \hbar\omega_0(\dot{N}_{\mathrm{in}} - \dot{N}_{\mathrm{out}})$$
$$\dot{\mathcal{U}}_f(t) = -\hbar\omega_L(\dot{N}_{\mathrm{in}} - \dot{N}_{\mathrm{out}})$$
$$\dot{\mathcal{V}}(t) = \hbar\delta(\dot{N}_{\mathrm{in}} - \dot{N}_{\mathrm{out}}), \tag{67}$$

which can be interpreted as a conversion of photons at energy $\hbar\omega_L$ into excitations of the atom at energy $\hbar\omega_0$. As already noticed in [44] and exploited in [85], this is the simultaneous conservation of energy and number of excitations in the atom-field system which implies a non-zero coupling energy flow proportional to the detuning $\delta$.

We can now compare the non-resonant energy balance captured by Eq. (66) to the one that would stem from a non-autonomous collision model picture of the OBEs. Collision models were first introduced in theory of open quantum system as effective reservoir models reproducing Lindblad open dynamics as OBEs [52–54]. They postulate the coupling of the atom to a reservoir of bosonic collision units, through an explicitly time-dependent Hamiltonian tuned such that the Lindblad master equation is retrieved upon tracing over the state of the units. The input units are prepared in the same displaced thermal states introduced in our autonomous collision model (See Sec. 3.2). The bare atomic Hamiltonian is still $H_S$,

the one of the reservoir of collision units is $H_f^{(\text{na})} = \hbar\omega_L \sum_n b_n^\dagger b_n$, and the time-dependent coupling reads

$$V_n^{(\text{na})}(t) = \begin{cases} i\sqrt{\frac{\gamma}{\Delta t}}(b_n\sigma_+ - b_n^\dagger\sigma_-), & t \in [t_n, t_{n+1}) \\ \\ 0 & \text{otherwise.} \end{cases} \tag{68}$$

Because of the explicit time-dependence, those models are intrinsically non-autonomous. A previously identified thermodynamic consequence of this non-autonomous nature is the existence of a work cost needed to switch on and off the interaction with each unit [54, 84]. Consequently, the energy balance of the non-autonomous collision model of the OBEs reads [54]

$$\dot{\mathcal{U}}_S(t) + \dot{\mathcal{U}}_f(t) = \dot{W}_{\text{switch}}, \tag{69}$$

where $\dot{W}_{\text{switch}}(t) \equiv \lim_{\gamma\Delta t \to 0} \left\langle \frac{\partial(\sum_n V_n^{(\text{na})}(t))}{\partial t} \right\rangle_t$ is the work to switch on and off the system-unit interaction. In the case of the OBEs, it is non-zero only out of resonance $\delta \neq 0$. Moreover, the internal energy flows of the system $\dot{\mathcal{U}}_S$ and the reservoir of collision units (field) $\frac{d}{dt}\left\langle H_f^{(\text{na})} \right\rangle_t \equiv \dot{\mathcal{U}}_f$ are identical by construction to those found for our autonomous model. By comparing Eq. (69) with Eq. (66), we find that $\dot{\mathcal{V}}(t) = -\dot{W}_{\text{switch}}$. Therefore, this is the variation of coupling energy in our autonomous model which provides the energy necessary to switch on and off the atomic coupling with the collision units.

The existence of a non-zero external work needed to generate the OBEs dynamics was a puzzling consequence of the non-autonomous collision models. Autonomous models show that external work is not actually necessary to generate the dynamics. In [86, 87], this issue was addressed by giving the units an extra motional degrees of freedom, whose kinetic energy was able to provide the necessary energy. In our case, no additional degree of freedom is required as the coupling term acts as a reservoir of energy.

# 9 Conclusions and outlooks

We have provided new elements to answer a long-lasting question in quantum thermodynamics: how can we measure heat-like and work-like quantities in the quantum realm, knowing that measuring a quantum system alters its thermodynamic balance because of measurement back-action? We have focused on optical Bloch equations, which are universal and widely used equations of quantum optics. We have built a new dynamical autonomous model which keeps track of atom-field correlations formed during a single collision, as well as the thermodynamic framework allowing us to characterize energy exchanges inside this autonomous atom-field system. This approach provides remarkably simple and operational expressions for work-like and the heat-like flows, which can be directly measured in the change of the mean field and of its fluctuations. These definitions have already been exploited in a handful of recently published works [31, 57, 59, 60, 77], both experimental and theoretical, based on heuristic arguments. In our manuscript, we root these definitions in a very general and thermodynamically consistent framework.

The difference between the standard (open) approach and our autonomous approach is the self-drive and its energetic counterpart, the self-work. We show that the self-work corresponds to some potentially reusable energy emitted in the bath, which is treated as heat in the open approach. Hence, the autonomous model carries the seeds for better energy

management at quantum scales - which translates into a tighter expression of the second law of thermodynamics.

More specifically, we expect the self-work to impact the optimization of quantum heat engines and the work cost of control operations at the fundamental level. In the first case, exploiting field displacements induced in the bath, that is re-using the self-work, can in principle lead to better performances (see e.g. [88] for a recent illustration in the case of an Otto engine at strong system-bath coupling). In the second, the negative self-work captures the unnecessary work expenditure beyond the minimum needed for a given control task, allowing to minimize it in view of reaching more energy-efficient control at quantum scales. An alternative strategy is the recycling of the driving pulse after its interaction with the atom. In this context, the degradation of the coherence of the pulse and consequently, the work it can provide on another atom is related to the entropy production $b\Sigma$. These considerations pertain to the emerging field of fundamental quantum energetics [89], with potential impact on the energetics of quantum technologies [39].

Interestingly, the constant sign of the self-work in our problem can be related to the sign of the self-reaction defined and analyzed in [79] from linear-response arguments. It reflects the impossibility for the thermal component of the input field to increase the coherence of the atomic dipole. This results is intimately related to the notion of passivity of the thermal state [90,91]. It would be instructive to analyze the radiation emitted by the atom in presence of incoherent initial field states which are non-passive i.e., in which the energy level occupancy would not be monotonously decreasing with the energy, to look for occurrences of positive self-work.

Finally, we stress that our findings are very general. On the dynamical side, the correlation tracking we have set up can be activated on all physical situations captured by an autonomous collision model. On the thermodynamic side, our definitions of work and heat flows can be used on all autonomous bipartite systems. Our framework provides a powerful new approach to analyze the thermodynamics of open systems, designed to take advantage of the tremendous technological progresses in the control and the engineering of environments.

# Acknowledgements

A.A. acknowledges the National Research Foundation, Singapore through the National Quantum Office, hosted in A*STAR, under its Centre for Quantum Technologies Funding Initiative (S24Q2d0009), the Plan France 2030 through the projects NISQ2LSQ (Grant ANR-22-PETQ-0006), OQuLus (Grant ANR-22-PETQ-0013), and OECQ, and the ANR Research Collaborative Project "Qu-DICE" (Grant ANR-PRC-CES47). M.M. acknowledges the support by Italian PNRR MUR project PE0000023-NQSTI. C.E. acknowledges funding from French National Research Agency (ANR) under grant ANR-22-CPJ1-0029-01. The authors are very grateful to Benjamin Huard as well as to the quantum energy team for inspiring discussions, especially Robert Whitney, Nicolò Piccione and Léa Bresque.

# A   Discrete field modes decomposition

In this Appendix, we discretize the field operator $B(x,t) = \sum_k g_k e^{-i\Delta\omega_k(t-x/v)} a_k$ present in the interaction $V(t)$ (see Eq. (13)), into $b_{n,j}$ modes satisfying $[b_{n,j}, b_{m,k}^\dagger] = \delta_{m,n}\delta_{j,k}$. This decomposition will be used in Appendix B to ensure that the atom interacts only with the input modes of light. Furthermore, we will relate these modes to the number and Hamiltonian

828    operators of the field that will be used to discretize the input field in Appendix C.
829       To obtain a discrete decomposition of the functions of time, we introduce the basis of
830    functions $w_{n,j}(t) = \frac{1}{\sqrt{\Delta t}}\Theta_n(t)e^{-i\frac{2\pi}{\Delta t}jt}$, for $n, j \in \mathbb{Z}$, where $\Theta_n(t) = 1$ for $t \in [t_n, t_{n+1}]$
831    and 0 everywhere else. The function $w_{n,j}$ has therefore only support on the time interval
832    $[t_n, t_{n+1}]$ and has an average frequency $2\pi j/\Delta t$. The basis is orthonormal in the sense that
833    $\langle w_{n,j}, w_{n',j'}\rangle = \int_{-\infty}^{\infty} dt w_{n,j}^*(t)w_{n',j'}(t) = \delta_{j,j'}\delta_{n,n'}$. We then have:

$$B(0, t) = \sum_{n,j} \sqrt{\gamma_j} w_{n,j}(t) b_{n,j}, \tag{A.1}$$

834    with $b_{n,j}$ the discrete field mode localized within time interval $[t_n, t_{n+1}]$ and with average
835    frequency $2\pi j/\Delta t$:

$$b_{n,j} = \frac{1}{\sqrt{\gamma_j \Delta t}} \int_{t_n}^{t_n+\Delta t} dt e^{i\frac{2\pi}{\Delta t}jt} B(0, t). \tag{A.2}$$

836    We have introduced the parameter $\gamma_j = 2\pi \sum_k |g_k|^2 \delta_D(\Delta\omega_k - \frac{2\pi j}{\Delta t})$. The $b_{n,j}$ are normalized
837    to verify:

$$\begin{aligned}
[b_{n,j}, b_{n',j'}^{\dagger}] &= \left[\int dt \sum_k g_k e^{-i\Delta\omega_k t} w_{n,j}(t) a_k, \int dt' \sum_{k'} g_{k'}^* e^{-i\Delta\omega_{k'}t'} w_{n',j'}^*(t') a_{k'}^{\dagger}\right] \\
&= \int dt \int dt' \sum_k |g_k|^2 e^{i\Delta\omega_k(t-t')} w_{n,j}(t) w_{n',j'}^*(t'). 
\end{aligned} \tag{A.3}$$

838    The correlation function of the field $C(t, t') = \sum_k |g_k|^2 e^{i\Delta\omega_k(t-t')}$ has a correlation time $\tau_c$.
839    Given the choice of coarse-graining time $\Delta t \gg \tau_c$, terms with $n \neq n'$ in Eq (A.3) will only
840    involve values of $|t - t'|$ much larger than $\tau_c$ and can then be neglected.

$$\begin{aligned}
[b_{n,j}(t), b_{n',j'}^{\dagger}(t)] &\simeq \frac{1}{\gamma_j}\frac{\delta_{n,n'}}{\Delta t} \int dt \Theta_n(t) \int dt' \Theta_n(t') C(t, t') e^{-\frac{2i\pi}{\Delta t}(jt-j't')} \\
&= \frac{1}{\gamma_j}\frac{\delta_{nn'}}{\Delta t} \int dt\, e^{-\frac{2i\pi}{\Delta t}(j-j')t}\Theta_n(t) \int d\tau \Theta_n(t-\tau)C(\tau, 0)e^{-\frac{2i\pi}{\Delta t}j'\tau} \\
&\simeq \delta_{n,n'} \frac{\int dt e^{-\frac{2i\pi}{\Delta t}(j-j')t}\Theta_n(t)}{\Delta t}\frac{\int_{-\infty}^{\infty} d\tau C(\tau, 0)e^{-\frac{2i\pi}{\Delta t}j'\tau}}{\gamma_j} \\
&= \delta_{n,n'}\delta_{j,j'}. 
\end{aligned} \tag{A.4}$$

841    To go to the second line we have changed integration variable from $t'$ to $\tau = t - t'$. To go to
842    the third line, we have used that $C(\tau, 0)$ is non zero for $|\tau| \lesssim \tau_c$ such that $\Theta(t-\tau)$ can be
843    approximated by $\Theta_n(t)$ in the integral over $\tau$.
844       Finally, it is useful to have in mind the correspondence relation that we derive from the
845    explicit expression of $B(0, t)$:

$$b_{n,j} = \sqrt{\frac{\Delta t}{\gamma_j}} \sum_k g_k e^{-i(\Delta\omega_k - \frac{2\pi j}{\Delta t})(t_n + \frac{\Delta t}{2})} \text{sinc}\left(\frac{1}{2}(\Delta\omega_k \Delta t - 2\pi j)\right) a_k. \tag{A.5}$$

846    This relation imply the equivalence of the two description in term of number of excitations

847  in the field:

$$
\begin{aligned}
\sum_{n,j} b_{n,j}^\dagger b_{n,j} &= \sum_{n,j} \frac{\Delta t}{\gamma_j} \sum_{k,l} g_k^* g_l a_k^\dagger a_l e^{i(\omega_l - \omega_k)(t_n + \Delta t)} \mathrm{sinc}\left(\frac{\Delta\omega_k \Delta t}{2} - \pi j\right) \mathrm{sinc}\left(\frac{\Delta\omega_l \Delta t}{2} - \pi j\right) \\
&\simeq \sum_j \frac{\Delta t}{\gamma_j} \sum_{k,l} g_k^* g_l a_k^\dagger a_l \frac{\delta_D(\omega_l - \omega_k)}{\Delta t} \mathrm{sinc}\left(\frac{\Delta\omega_k \Delta t}{2} - \pi j\right) \mathrm{sinc}\left(\frac{\Delta\omega_l \Delta t}{2} - \pi j\right) \\
&= \sum_j \frac{1}{\gamma_j} \sum_k |g_k|^2 a_k^\dagger a_k \mathrm{sinc}^2\left(\frac{\Delta\omega_k \Delta t}{2} - \pi j\right) \sum_l \delta_D(\omega_l - \omega_k) \\
&\simeq \sum_j \sum_k a_k^\dagger a_k \mathrm{sinc}^2\left(\frac{\Delta\omega_k \Delta t}{2} - \pi j\right) \\
&= \sum_k a_k^\dagger a_k .
\end{aligned}
\tag{A.6}
$$

848  To go to the fourth line, we have used that the sinc functions are peaked around
849  $\Delta\omega_k = 2\pi j/\Delta t$, such that $|g_k|^2 \sum_l \delta_D(\omega_k - \omega_l) \simeq \gamma_j$ provided the coupling coefficients
850  can be considered constant over the frequency interval $2\pi/\Delta t$, which holds true in the
851  limit $\Delta t \gg \tau_c$ considered in this article. To go to the last line, we have used the identity
852  $\sum_{j=-\infty}^{\infty} \mathrm{sinc}^2(x - \pi j) = 1$.
853  Using that $2\pi/\Delta t \ll \omega_k$ over the frequency range in which the atom and field interact, we
854  can also deduce the equivalence relation in terms of the field Hamiltonian:

$$
\begin{aligned}
\sum_{n,j}\left(\omega_0 + \frac{2\pi j}{\Delta t}\right) b_{n,j}^\dagger b_{n,j} &\simeq \sum_j \left(\omega_0 + \frac{2\pi j}{\Delta t}\right) \sum_k a_k^\dagger a_k \mathrm{sinc}^2\left(\omega_0 + \frac{2\pi j}{\Delta t}\right) \\
&\simeq \sum_j \sum_k \omega_k a_k^\dagger a_k \mathrm{sinc}^2\left(\frac{\Delta\omega_k \Delta t}{2} - \pi j\right) \\
&= \sum_k \omega_k a_k^\dagger a_k .
\end{aligned}
\tag{A.7}
$$

## B  Coarse-grained unitary evolution operator

856  Here, we discretize the evolution into time intervals of $\Delta t$ using the Magnus expansion in
857  order to obtain Eq. (17) and find the order of error involved. To do so, we derive the explicit
858  expression of the Unitary evolution operator over a time interval $\Delta t$, whose formal expression
859  reads

$$
\mathcal{U}(t_n, t_{n+1}) = \mathcal{T} e^{-i/\hbar \int_{t_n}^{t_{n+1}} dt\, V(t)}
\tag{B.1}
$$

860  Magnus expansion allows one to remove the time-ordering $\mathcal{U}_n = e^{\Omega(t_n, t_{n+1})}$ with

$$
\begin{aligned}
\Omega(t_n, t_{n+1}) &= -\frac{i}{\hbar} \int_{t_n}^{t_{n+1}} dt\, V(t) - \frac{1}{2\hbar^2} \int_{t_n}^{t_{n+1}} dt \int_{t_n}^{t} dt' \left[V(t), V(t')\right] + \dots \\
&= \Omega_1 + \Omega_2 .
\end{aligned}
\tag{B.2}
$$

861  We now use the decomposition Eq.(A.1) in discrete modes $b_{n,j}$ to evaluate $\Omega(t_n, t_{n+1})$. We
862  first note that:

$$
\begin{aligned}
\Omega_1 &= \int_{t_n}^{t_n + \Delta t} dt\, B(0, t)\sigma_+ + \text{H.c.} \\
&= \sqrt{\gamma \Delta t}\left(b_{n,0}\sigma_+ - b_{n,0}^\dagger \sigma_-\right).
\end{aligned}
\tag{B.3}
$$

On the other hand:

$$
\begin{aligned}
\Omega_2 &= -\frac{1}{2}\int_{t_n}^{t_n+\Delta t} dt \int_{t_n}^{t} dt' \left(B(0,t)B^\dagger(0,t')\sigma_+\sigma_- + B^\dagger(0,t)B(0,t')\sigma_-\sigma_+ - \text{H.c.}\right) \\
&= -\frac{1}{2}\int_{t_n}^{t_n+\Delta t} dt \int_{t_n}^{t} dt' \sum_{j'} \sqrt{\frac{\gamma_{j'}}{\Delta t}} \left(e^{i\frac{2\pi}{\Delta t}j't'}B(0,t)b_{n,j'}^\dagger\sigma_+\sigma_- + e^{-i\frac{2\pi}{\Delta t}j't'}B^\dagger(0,t)b_{n,j'}\sigma_-\sigma_+ - \text{H.c.}\right) \\
&= -\frac{1}{2}\sum_{j'\neq 0}\sqrt{\frac{\gamma_{j'}}{\Delta t}}\int_{t_n}^{t_n+\Delta t} dt\, \frac{\Delta t}{i2\pi j'}\left[\left(e^{i\frac{2\pi}{\Delta t}j't}-1\right)B(0,t)b_{n,j'}^\dagger\sigma_+\sigma_- - \left(e^{-i\frac{2\pi}{\Delta t}j't}-1\right)B^\dagger(0,t)b_{n,j'}\sigma_-\sigma_+\right] - \text{H.c.} \\
&\quad -\frac{1}{2}\sqrt{\frac{\gamma}{\Delta t}}\int_{t_n}^{t_{n+1}} dt\, \left((t-t_n)B(0,t)\,b_{n,0}^\dagger\sigma_+\sigma_- + (t-t_n)B^\dagger(0,t)\,b_{n,0}\sigma_-\sigma_+ - \text{H.c.}\right).
\end{aligned}
\tag{B.4}
$$

To go to the second line, we have introduced the discrete mode decomposition of $B(0,t')$ for $t' \in [t_n, t_{n+1}]$ (which corresponds to keeping only the terms with time label $n$). Noting that the integration over $t$ yields by definition (see Eq. (A.2)) the modes $b_{n,j'}$ up to a factor $\sqrt{\gamma_{j'}\Delta t}$, we obtain:

$$
\begin{aligned}
\Omega_2 &= -\frac{\Delta t}{4\pi}\sum_{j'\neq 0}\left[\frac{\sqrt{\gamma_{j'}}}{ij'}\left[\sqrt{\gamma_{j'}}\left(b_{n,j'}^\dagger b_{n,j'}\sigma_z + \sigma_+\sigma_-\right) + \sqrt{\gamma}\left(b_{n,0}^\dagger b_{n,j'}\sigma_-\sigma_+ - b_{n,0}b_{n,j'}^\dagger\sigma_+\sigma_-\right)\right] - \text{H.c.}\right] \\
&\quad -\sum_{j\neq 0}\frac{i\sqrt{\gamma\gamma_{j'}}\Delta t}{4\pi j'}\left(b_{n,0}^\dagger b_{n,j'} + b_{n,j'}^\dagger b_{n,0}\right)\sigma_z \\
&= \frac{i\Delta t}{2\pi}\sum_{j'\neq 0}\left(\frac{\gamma_{j'}}{j'}\left(b_{n,j'}^\dagger b_{n,j'}\sigma_z + \sigma_+\sigma_-\right) - \frac{\sqrt{\gamma\gamma_{j'}}}{j'}\left(b_{n,0}^\dagger b_{n,j'} + b_{n,j'}^\dagger b_{n,0}\right)\sigma_z\right)
\end{aligned}
\tag{B.5}
$$

Here, we have chosen $t_0 = 0$ without loss of generality. We see that $\Omega_2$ contains the term: $\Omega_{2,1} = -\hbar\delta_{\text{LS}}\sigma_+\sigma_- + \frac{i\Delta t}{2\pi}\sum_{j'\neq 0}\frac{\gamma_{j'}}{j'}b_{n,j'}^\dagger b_{n,j'}\sigma_z$, with $\delta_{\text{LS}} = \frac{1}{2\pi}\sum_{j'>0}\frac{\gamma_{j'}}{j'}$. $\delta_{\text{LS}}$ induces a correction to the atom frequency (Lamb shift) while there is also a dispersive coupling with the higher frequency modes ($j \neq 0$) which causes a shift of the atom's frequency (light shift) that we can evaluate by taking the average over a thermal state of the field in each time bin. We find that, it's magnitude is

$$
\begin{aligned}
\Delta t\,\delta_{ls} = \left|\frac{i\Delta t}{2\pi}\sum_{j\neq 0}\frac{\gamma_j\left\langle b_{n,j}^\dagger b_{n,j}\right\rangle}{j}\right| &\lesssim \Delta t\gamma\left|\int_{\frac{\pi}{\Delta t}}^{\Delta_{bw}/2} d\omega\,\frac{n[\omega_0+\omega]-n[\omega_0-\omega]}{\omega}\right| \\
&\simeq \Delta t\gamma\left|\int_{\frac{\pi}{\Delta t}}^{\Delta_{bw}/2} d\omega\,\frac{2\bar{n}_{\text{th}}(\bar{n}_{\text{th}}+1)\beta\omega}{\omega}\right| \\
&\simeq \Delta t\,\bar{n}_{\text{th}}(\bar{n}_{\text{th}}+1)\gamma\beta\Delta_{\text{bw}}
\end{aligned}
\tag{B.6}
$$

where $\Delta_{bw}$ is the band-width of the atom-field coupling (see main text) and we have approximated the sum over $j$ with an integral over frequencies $\omega \equiv \omega_j = \frac{2\pi j}{\Delta t}$. In the first line, we have introduced $n[\omega] = (e^{-\beta\hbar\omega}-1)^{-1}$ which is the Bose-Einstein distribution of the field, and to go to the second line we have assumed that the bandwidth $\Delta_{\text{bw}}$ is much smaller than $\omega_0$ so that we can expand the distribution around $\omega_0$. One can notice that $\delta_{ls} \ll \omega_0$ from the fact that $\tau_c \geq \hbar\beta, \Delta_{\text{bw}}^{-1}$ [23] and $\gamma\tau_c \ll 1$ (and similarly for $\delta_{LS}$).

One can recover the usual expression to the Lamb shift (appearing e.g. in the context of

the derivation of the OBEs) by taking into account the limit $\Delta t \gg \tau_c \geq \Delta_{\mathrm{bw}}^{-1}, \hbar\beta$:

$$
\begin{aligned}
\delta_{\mathrm{LS}} &= \frac{1}{2\pi} \sum_{j'>0} \frac{1}{j'} \sum_k |g_k|^2 \delta_D\left(\Delta\omega_k - \frac{2\pi j'}{\Delta t}\right) \\
&\simeq \frac{1}{2\pi} \mathrm{P} \int d\omega \sum_k |g_k|^2 \frac{\delta_D(\Delta\omega_k - \omega)}{\omega} \\
&= \frac{1}{2\pi} \sum_{k \neq 0} \frac{|g_k|^2}{\Delta\omega_k},
\end{aligned} \tag{B.7}
$$

where P stands for Cauchy's principal value and again we have approximated the sum over $j'$ with an integral over frequencies $\omega \equiv \omega_j$.

The other terms of $\Omega_2$, induces an effective coupling, mediated by the atom, between the modes $b_{n,j}$ for $j = 0$ (the closer to the atom's frequency) to modes further detuned from the atom $j \neq 0$. This coupling term is already of first order in $\gamma\Delta t$. Assuming a factorized initial thermal state of the modes $b_{n,j}$ (and no initial correlation with the atom) ensures that $\langle b_{n,j}(0)\rangle = 0$, such that this coupling term yield no contribution at first order to the reduced dynamics of the atom or of the $b_{n,j}$ modes.

Reabsorbing the Lamb shift in the definition of $\omega_0$, we can therefore approximate $\Omega(t_n, t_{n+1}) = \sqrt{\gamma\Delta t}\left(b_{n,0}\sigma_+ + b_{n,0}^\dagger\sigma_-\right) + o(\gamma\Delta t)$. To first order in $\gamma\Delta t$, the atom therefore effectively interacts only with the modes $b_{n,j=0} \equiv b_n$. To be consistent with our first order truncation of $\Omega(t_n, t_{n+1})$, we finally expand the exponential in $\mathcal{U}(t_n, t_{n+1})$ to obtain

$$
\mathcal{U}(t_n, t_{n+1}) = 1 + \sqrt{\gamma\Delta t}\left(b_n\sigma_+ - b_n^\dagger\sigma_-\right) + \frac{\gamma\Delta t}{2}\left(b_n\sigma_+ - b_n^\dagger\sigma_-\right)^2 + o(\gamma\Delta t). \tag{B.8}
$$

# C   Decomposition of a displaced-Thermal field in temporal bins

Here we discretize a flat and continuous displaced-thermal field into collision units and also relate the Rabi frequency to the amplitude of these units. Starting with a thermal field, it is described by the density matrix $\rho_f^\beta = \exp\{-\beta H_f\}/Z$ as defined in the main text. Using the equivalence relation Eq. (A.7), we see that such state is well approximated by a factorized of the modes $b_{n,j}$, namely: $\rho_f^\beta \simeq \bigotimes_{n,j} e^{-\beta\left(\omega_0 + \frac{2\pi j}{\Delta t}\right)b_{n,j}^\dagger b_{n,j}}/Z$. In particular, the modes $b_n \equiv b_{n,j=0}$ involved in the collision model description of the dynamics are described by the state:

$$
\rho_f^{\beta,(j=0)} = \bigotimes_n \frac{e^{-\beta\omega_0 b_n^\dagger b_n}}{Z_0} = \bigotimes_n \eta_n^\beta, \tag{C.1}
$$

with $Z_0 = \mathrm{Tr}\{e^{-\beta\omega_0 \sum_n b_n^\dagger b_n}\}$.

Let the field be displaced by an amplitude $\alpha_L$ at frequency $\omega_L$. The displacement operator has the form $\mathcal{D}(\alpha_L) = \exp\{\alpha_L a_L^\dagger - \alpha_L^* a_L\}$ in terms of the operator $a_L$ destroying excitations of the field's mode of frequency $\omega_L$. Eq. (A.5) implies that all the modes $b_n$ are displaced by an amplitude $\alpha_n$ which verifies:

$$
\alpha_n = \sqrt{\frac{\Delta t}{\gamma}} g_L \alpha_L \mathrm{sinc}\left((\omega_L - \omega_0)\frac{\Delta t}{2}\right) e^{-i(\omega_L - \omega_0)(t_n + \frac{\Delta t}{2})}, \tag{C.2}
$$

$$
|\alpha_L| = \lim_{\gamma\Delta t \to 0} \sqrt{\frac{\gamma}{\Delta t}} \frac{|\alpha_n|}{g_L}. \tag{C.3}
$$

The displaced-thermal field can hence be written as,

$$\rho_f^{(j=0)}(t_0) = \bigotimes_n \mathcal{D}(\alpha_n) \eta_n^\beta \mathcal{D}^\dagger(\alpha_n), \tag{C.4}$$

where, $t_0 = 0$ is the initial time before the interaction. Hence we find,

$$\sqrt{\gamma}\,\langle b_{\text{in}}(t)\rangle_t = \lim_{\gamma\Delta t \to 0} \sqrt{\frac{\gamma}{\Delta t}}\,\langle b_{x_0}(t_n)\rangle_{t_n} = \lim_{\gamma\Delta t \to 0} \sqrt{\frac{\gamma}{\Delta t}}\,\alpha_n = g_L \alpha_L e^{-i(\omega_L - \omega_0)t} = e^{-i(\omega_L - \omega_0)t}\Omega/2. \tag{C.5}$$

# D   Field's reduced dynamics

In this section, we analyze the effect of the different orders of evolution of each collision unit of the field during its interaction with the atom, on its amplitude and excitations number. In particular, we derive Eq. (22), Eq. (26), Eq. (27), Eq. (37) and Eq. (38). We begin by deriving the input-output equation Eq. (22). Taking the change in the coherent part of the field over the collision with the atom, i.e., $\langle b_{x_1}(t_{n+1})\rangle_{t_{n+1}} - \langle b_{x_0}(t_n)\rangle_{t_n} = \text{Tr}\{b_n \Delta_n \rho_f\}$ and inject that $\mathcal{H}_f(t_n) \equiv \langle V_n\rangle_S/\Delta t = i\sqrt{\gamma/\Delta t}(\langle\sigma_+\rangle_{t_n} b_n - \langle\sigma_-\rangle_{t_n} b_n^\dagger)$. We find

$$\Delta_n \rho_f^{(1)} = \text{Tr}_S\{\Delta_n \rho^{(1)}\} = -i\Delta t\left[\mathcal{H}_f(t_n), \rho_f(t_n)\right], \tag{D.1}$$

$$\text{Tr}\{b_n \Delta_n \rho_f^{(1)}\} = -i\Delta t\,\text{Tr}\{b_n\left[\mathcal{H}_f(t_n), \rho_f(t_n)\right]\} = -\sqrt{\gamma\Delta t}\,\langle\sigma_-\rangle_{t_n}, \tag{D.2}$$

while, we find $\text{Tr}\{b_{x_0}(t_n)\Delta_n \rho_f^{(2)}\} = \mathcal{O}(\Delta t\sqrt{\Delta t})$ as $\Delta t \langle b_n\rangle_{t_n} = o(\Delta t\sqrt{\Delta t})$. Hence, only the first order term of the Dyson series provides a non-negligible contribution to the change in the coherent part of the field. These results are equivalent to the input-output equations [69] and in the continuous time limit leads to Eq. (22) as shown in Section 3.3. The first order term also displays photons that are emitted through to stimulated emission $N_{\text{drive}}$. We find:

$$\text{Tr}\{b_n^\dagger b_n \Delta_n \rho_f^{(1)}\} = -i\Delta t\,\text{Tr}\{b_n^\dagger b_n\left[\mathcal{H}_f(t_n), \rho_f(t_n)\right]\}$$
$$= -\sqrt{\gamma\Delta t}(\langle\sigma_+\rangle_{t_n}\langle b_{x_0}(t_n)\rangle_{t_n} + \langle\sigma_-\rangle_{t_n}\langle b_{x_0}^\dagger(t_n)\rangle_{t_n}). \tag{D.3}$$

In the continuous limit, this leads to Eq. (26).

To compute the effect on the incoherent part of the field, we start from Eqs. (30) in the case of the field $f$ (taking the trace over the atom), and inject $\langle V_n\rangle_S$. We first focus on the term $\Delta_n \rho_f^{(2,f)}$. Unlike the case of the atom, this term is of order $\mathcal{O}(\sqrt{\gamma\Delta t})$, such that

$$\begin{aligned}
\Delta_n \rho_f^{(2,f)} &= -\frac{i\Delta t}{2}[\mathcal{H}_f(t_n), \Delta_n \rho_f^{(1)}] \\
&= -\frac{\Delta t^2}{2}[\mathcal{H}_f(t_n), [\mathcal{H}_f(t_n), \rho_f(t_n)]] \\
&= \frac{\gamma\Delta t}{2}[b_n\langle\sigma_+\rangle_{t_n} - b_n^\dagger\langle\sigma_-\rangle_{t_n}, [b_n\langle\sigma_+\rangle_{t_n} - b_n^\dagger\langle\sigma_-\rangle_{t_n}, \rho_f(t_n)]]
\end{aligned} \tag{D.4}$$

is of first order in $\gamma\Delta t$. This terms is in particular important to compute the change of field photon number across a collision (see main text).

The self-drive Hamiltonian for the field takes the explicit form:

$$\mathcal{H}_f^s(t_n) = -\frac{i\gamma}{2}\langle\sigma_z\rangle_{t_n}\left(\langle b_n^\dagger\rangle_{t_n} b_n - \langle b_n\rangle_{t_n} b_n^\dagger\right) \tag{D.5}$$

As $\langle b_n \rangle_{t_n} = \mathcal{O}(\Omega \sqrt{\frac{\Delta t}{\gamma}})$ (see Appendix C), we see that the term $\Delta_n \rho_f^{(2,S)}$ is of order $\mathcal{O}(\Delta t^{3/2})$ and hence the field's self-drive is negligible in the continuous limit.

Now we use $\Delta_n \rho_f^{(2,f)}$ to derive its contribution to the change in the photon number of the field in Eq. (36). This is given by:

$$
\begin{aligned}
\mathrm{Tr}\left\{ b_n^\dagger b_n \Delta_n \rho_f^{(2,f)} \right\} = & \frac{\gamma \Delta t}{2} \mathrm{Tr}\left\{ \left[\left[ b_n^\dagger b_n, \left( b_n \langle \sigma_+ \rangle_{t_n} - b_n^\dagger \langle \sigma_- \rangle_{t_n} \right)\right], \left( b_n \langle \sigma_+ \rangle_{t_n} - b_n^\dagger \langle \sigma_- \rangle_{t_n} \right)\right] \rho_f(t_n) \right\} \\
= & \gamma \Delta t \left| \langle \sigma_- \rangle_{t_n} \right|^2 ,
\end{aligned}
\tag{D.6}
$$

which contributes to the change in the field's amplitude. We can now compute the second order correlation for the field, i.e., $\Delta_n \rho_f^{(2,\chi)}$ by taking the trace over the atom in $\Delta_n \rho^{(2,\chi)}$ (see Eqs. (30)). Substituting $\langle V_n \rangle_S$ and $\langle V_n \rangle_f$ we find:

$$
\begin{aligned}
\Delta_n \rho_f^{(2,\chi)} = & -\frac{1}{2} \mathrm{Tr}_S \left\{ \left[ V_n, \left[ V_n - \langle V_n \rangle_f - \langle V_n \rangle_S, \rho_n \right]\right]\right\} \\
= & -\frac{1}{2} \mathrm{Tr}_S \left\{ \left[ V_n, \right. \right. \\
& \left. \left. \left[ V_n - i\sqrt{\gamma \Delta t} \left( \left( \langle b_n \rangle_{t_n} \sigma_+ - \langle b_n^\dagger \rangle_{t_n} \sigma_- \right) + \left( \langle \sigma_+ \rangle_{t_n} b_n - \langle \sigma_- \rangle_{t_n} b_n^\dagger \right) \right), \rho(t_n) \right]\right]\right\}
\end{aligned}
\tag{D.7}
$$

Its contribution to the change in the photon number of the field (Eq. (36)) is given by:

$$
\begin{aligned}
& \mathrm{Tr}\{ b_n^\dagger b_n \Delta_n \rho_f^{(2,\chi)} \} \\
= & -\frac{1}{2} \mathrm{Tr}\left\{ \left[ \left[ b_n^\dagger b_n, V_n \right], \right. \right. \\
& \left. \left. V_n - i\sqrt{\gamma \Delta t} \left( \left( \langle b_n \rangle_{t_n} \sigma_+ - \langle b_n^\dagger \rangle_{t_n} \sigma_- \right) + \left( \langle \sigma_+ \rangle_{t_n} b_n - \langle \sigma_- \rangle_{t_n} b_n^\dagger \right) \right) \right] \rho(t_n) \right\} \\
= & \frac{\gamma \Delta t}{2} \mathrm{Tr}\left\{ \left[ -i\sqrt{\gamma \Delta t} \left( \sigma_+ b_n + \sigma_- b_n^\dagger \right), \right. \right. \\
& \left. \left. V_n - i\sqrt{\gamma \Delta t} \left( \left( \langle b_n \rangle_{t_n} \sigma_+ - \langle b_n^\dagger \rangle_{t_n} \sigma_- \right) + \left( \langle \sigma_+ \rangle_{t_n} b_n - \langle \sigma_- \rangle_{t_n} b_n^\dagger \right) \right) \right] \rho(t_n) \right\} \\
= & \gamma \Delta t \left( \left( \bar{n}_{\mathrm{th}} + \frac{1}{2} \right) \langle \sigma_z \rangle_{t_n} + \frac{1}{2} \right) - \gamma \Delta t \left| \langle \sigma_- \rangle_{t_n} \right|^2 ,
\end{aligned}
\tag{D.8}
$$

which governs the fluctuations of the photon number change and in the continuous limit gives Eq. (38). Using $\mathrm{Tr}\{ b_n^\dagger b_n \Delta_n \rho_f^{(2)} \} = \mathrm{Tr}\{ b_n^\dagger b_n \Delta_n \rho_f^{(2,f)} \} + \mathrm{Tr}\{ b_n^\dagger b_n \Delta_n \rho_f^{(2,\chi)} \}$ and $\mathrm{Tr}\left\{ b_n^\dagger b_n \Delta_n \rho_f^\otimes \right\} = \mathrm{Tr}\left\{ b_n^\dagger b_n \Delta_n \rho_f^{(1)} \right\} + \mathrm{Tr}\left\{ b_n^\dagger b_n \Delta_n \rho_f^{(2,f)} \right\}$ we find Eq. (27) and Eq. (37) in the continuous limit respectively.

# E  Atom's reduced dynamics

Here, we will evaluate the second order contributions to the unitary part of the atomic dynamics. Let us begin with the self-driving term $\Delta_n \rho_S^{(2,f)}$. Substituting $V_n$ and $\langle V_n \rangle_S$ in

Eq. (30) and taking the trace with the field, we find,

$$
\begin{aligned}
\Delta_n \rho_S^{(2,f)} &= \mathrm{Tr}_f \left\{ \Delta_n \rho^{(2,f)} \right\} \\
&= -\frac{1}{2} \mathrm{Tr}_f \left\{ \left[ V_n, \left[ \langle V_n \rangle_S, \rho_f(t_n) \right] \rho_S(t_n) \right] \right\} \\
&= -\frac{1}{2} \mathrm{Tr}_f \left\{ V_n \left[ \langle V_n \rangle_S, \rho_f(t_n) \right] \rho_S(t_n) \right\} + \frac{1}{2} \mathrm{Tr}_f \left\{ \left[ \langle V_n \rangle_S, \rho_f(t_n) \right] \rho_S(t_n) V_n \right\} \\
&= -\frac{i}{\hbar} \Delta t \left[ -i\hbar \frac{\gamma}{2} \left( \langle \sigma_- \rangle_{t_n} \sigma_+ - \langle \sigma_+ \rangle_{t_n} \sigma_- \right), \rho_S(t_n) \right] \\
&= -\frac{i}{\hbar} \Delta t \left[ \mathcal{H}_S^s(t_n), \rho_S(t_n) \right],
\end{aligned} \tag{E.1}
$$

which is a driving term resulting from the change of the field state during the collision. This reciprocal driving leads to self-driving of the atom. Similarly substituting $\langle V_n \rangle_f$ in Eq. (30), we find that $\Delta_n \rho_S^{(2,S)}$ is

$$
\begin{aligned}
\Delta_n \rho_S^{(2,S)} &= \mathrm{Tr}_f \left\{ \Delta_n \rho^{(2,S)} \right\} \\
&= -\frac{1}{2} \left[ \langle V_n \rangle_f, \left[ \langle V_n \rangle_f, \rho_S(t_n) \right] \right] \\
&= \frac{\gamma \Delta t}{2} \left[ \langle b_n \rangle_{t_n} \sigma_+, \left[ \left( \langle b_n \rangle_{t_n} \sigma_+ - \langle b_n^\dagger \rangle_{t_n} \sigma_- \right), \rho_S(t_n) \right] \right] \\
&\quad - \frac{\gamma \Delta t}{2} \left[ \langle b_n^\dagger \rangle_{t_n} \sigma_-, \left[ \left( \langle b_n \rangle_{t_n} \sigma_+ - \langle b_n^\dagger \rangle_{t_n} \sigma_- \right), \rho_S(t_n) \right] \right].
\end{aligned} \tag{E.2}
$$

As $\Delta t \left| \langle b_n \rangle_{t_n} \right|^2 = \frac{1}{\gamma} \mathcal{O}\left( \Omega^2 \Delta t^2 \right)$, this term is negligible and does not contribute to the atomic dynamics.

# F  Energy Analysis and Expressions

In this section, we derive relations between the energy flows and express them in terms of the atom and field averages in the regime of the OBEs.

## F.1  Expressions of energetic quantities in terms of atom averages for initial coherent-thermal field

Here, we compute the rate of change (fluxes) of the BQE quantities in terms of the averages of atomic operators and list them for convenience. To do so we use the definitions introduced in section 5 along with the balance equations and substitute the OBEs (Eq. (1)). This gives the change in the energies over the collision interval. We also introduce the approximation to relate the atom and field energetic quantities. The BQE quantities are

958 • Atom internal energy flow:

$$
\begin{aligned}
\frac{\Delta_n \mathcal{U}_S(t_n)}{\Delta t} =& \text{Tr}\left\{ H_S \frac{\Delta_n \rho_S}{\Delta t} \right\} \\
=& -i\frac{\omega_0}{2}\text{Tr}\{[\sigma_z, H_D(t_n)]\rho_S(t_n)\} \\
&+ \frac{\hbar\omega_0 \gamma \bar{n}_{\text{th}}}{2}\text{Tr}\left( \sigma_z \sigma_+ \rho_S(t_n)\sigma_- - \frac{1}{2}\sigma_z\{\sigma_-\sigma_+, \rho_S(t_n)\} \right) \\
&+ \frac{\gamma\hbar\omega_0(\bar{n}_{\text{th}}+1)}{2}\text{Tr}\left( \sigma_z \sigma_- \rho_S(t_n)\sigma_+ - \frac{1}{2}\sigma_z\{\sigma_+\sigma_-, \rho_S(t_n)\} \right) \\
=& \frac{\hbar\omega_0\Omega}{2}\left( \langle \sigma_+ e^{-i(\omega_L-\omega_0)t_n} \rangle_{t_n} + \text{H.c.} \right) - \gamma\hbar\omega_0\left( \bar{n}_{\text{th}} + \frac{1}{2} \right)\langle\sigma_z\rangle_{t_n} - \frac{\gamma\hbar\omega_0}{2}.
\end{aligned}
$$
(F.1)

959 • Coupling energy flow (see Appendix F.4):

$$
\begin{aligned}
\frac{\Delta_n \mathcal{V}(t_n)}{\Delta t} =& \frac{\Delta_n \text{Tr}\{H_D(t_n)\rho_S(t_n)\}}{\Delta t} \\
=& \frac{i\hbar\Omega}{2}\frac{\Delta_n \text{Tr}\left\{ \left( e^{-i(\omega_L-\omega_0)t_n}\sigma_+ - e^{i(\omega_L-\omega_0)t_n}\sigma_- \right)\rho_S(t_n) \right\}}{\Delta t} \\
=& \frac{i\hbar\Omega}{2}\text{Tr}\left\{ \rho_S(t_n)\frac{\Delta_n\left( e^{-i(\omega_L-\omega_0)t_n}\sigma_+ - e^{i(\omega_L-\omega_0)t_n}\sigma_- \right)}{\Delta t} \right\} \\
&+ \frac{i\hbar\Omega}{2}\text{Tr}\left\{ \left( e^{-i(\omega_L-\omega_0)t_n}\sigma_+ - e^{i(\omega_L-\omega_0)t_n}\sigma_- \right)\frac{\Delta_n \rho_S}{\Delta t} \right\} \\
=& \frac{\hbar\Omega(\omega_L-\omega_0)}{2}\left( e^{-i(\omega_L-\omega_0)t_n}\langle\sigma_+\rangle_{t_n} + e^{i(\omega_L-\omega_0)t_n}\langle\sigma_-\rangle_{t_n} \right) - \gamma\left( \bar{n}_{\text{th}} + \frac{1}{2} \right)\langle H_D(t_n)\rangle_{t_n}.
\end{aligned}
$$
(F.2)

960 • Field internal energy flow:

$$
\begin{aligned}
\frac{\Delta_n \mathcal{U}_f(t_n)}{\Delta t} =& -\frac{\Delta_n \mathcal{U}_S(t_n)}{\Delta t} - \frac{\Delta_n \mathcal{V}(t_n)}{\Delta t} \\
=& -\frac{\hbar\omega_0\Omega}{2}\left( e^{-i(\omega_L-\omega_0)t_n}\langle\sigma_+\rangle_{t_n} + \text{H.c.} \right) + \hbar\omega_0\gamma\left( \bar{n}_{\text{th}} + \frac{1}{2} \right)\langle\sigma_z\rangle_{t_n} + \frac{\gamma\hbar\omega_0}{2} \\
&- \frac{\hbar\Omega(\omega_L-\omega_0)}{2}\left( e^{-i(\omega_L-\omega_0)t_n}\langle\sigma_+\rangle_{t_n} + \text{H.c.} \right) + \gamma\left( \bar{n}_{\text{th}} + \frac{1}{2} \right)\langle H_D(t_n)\rangle_{t_n} \\
=& -\frac{\hbar\omega_L\Omega}{2}\left( e^{-i(\omega_L-\omega_0)t_n}\langle\sigma_+\rangle_{t_n} + \text{H.c.} \right) + \gamma\left( \bar{n}_{\text{th}} + \frac{1}{2} \right)\left( \hbar\omega_0\langle\sigma_z\rangle_{t_n} \right. \\
&\left. + \langle H_D(t_n)\rangle_{t_n} \right) + \frac{\gamma\hbar\omega_0}{2}.
\end{aligned}
$$
(F.3)

961 From the last lines of Eq. (F.1) and Eq. (F.3), we notice that $\Delta\mathcal{U}_S/\Delta t = -(\hbar\omega_0/\hbar\omega_L)\Delta\mathcal{U}_f/\Delta t$
962 up to a relative error of $\mathcal{O}(\Omega/\omega_0, (\omega_L-\omega_0)/\omega_0)$. This relative error is negligible in the regime
963 of validity of the OBEs.

964 • Atom bipartite work flow (using results of Appendix E):

$$
\begin{aligned}
\frac{\Delta_n[b\mathcal{W}_S](t_n)}{\Delta t} &= \mathrm{Tr}_S\left\{H_S\frac{\mathrm{Tr}_f\{\Delta_n\rho^\otimes\}}{\Delta t}\right\} \\
&= \mathrm{Tr}\left\{H_S\frac{\Delta_n\rho_S^{(1)}}{\Delta t}\right\} + \mathrm{Tr}\left\{H_S\frac{\Delta_n\rho_S^{(2,f)}}{\Delta t}\right\} \\
&= \frac{\hbar\omega_0\Omega}{2}\left(e^{-i(\omega_L-\omega_0)t_n}\langle\sigma_+\rangle_{t_n} + e^{i(\omega_L-\omega_0)t_n}\langle\sigma_-\rangle_{t_n}\right) - \gamma\hbar\omega_0\left|\langle\sigma_-\rangle_{t_n}\right|^2.
\end{aligned}
$$
(F.4)

965 • Field bipartite work flow (see Appendix G for the derivation of the first line of the
966 following):

$$
\begin{aligned}
\frac{\Delta_n[b\mathcal{W}_f](t_n)}{\Delta t} &= \mathrm{Tr}_f\left\{H_f\frac{\mathrm{Tr}_S\{\Delta_n\rho^\otimes\}}{\Delta t}\right\} = -\frac{\Delta_n W(t_n)}{\Delta t} - \frac{\Delta_n W_{\mathrm{self}}(t_n)}{\Delta t} \\
&= -\mathrm{Tr}\left\{\rho_S(t_n)\frac{\Delta_n H_D(t_n)}{\Delta t}\right\} - \mathrm{Tr}\left\{\rho_S(t_n)\frac{\Delta_n\mathcal{H}_S^s(t_n)}{\Delta t}\right\} \\
&\quad + \frac{i}{\hbar}\mathrm{Tr}\{\rho_S(t_n)[H_S,H_D(t_n)]\} + \frac{i}{\hbar}\mathrm{Tr}\{\rho_S(t_n)[H_S,\mathcal{H}_S^s(t_n)]\} \\
&= -\frac{\hbar\omega_L\Omega}{2}\left(e^{-i(\omega_L-\omega_0)t_n}\langle\sigma_+\rangle_{t_n} + e^{i(\omega_L-\omega_0)t_n}\langle\sigma_-\rangle_{t_n}\right) - \frac{\gamma}{2}\langle H_D(t_n)\rangle_{t_n}\langle\sigma_z\rangle_{t_n} \\
&\quad + \gamma\hbar\omega_0\left|\langle\sigma_-\rangle_{t_n}\right|^2.
\end{aligned}
$$
(F.5)

967 • Atom bipartite heat flow:

$$
\begin{aligned}
\frac{\Delta_n[b\mathcal{Q}_S](t_n)}{\Delta t} &= \frac{\Delta_n\mathcal{U}_S(t_n)}{\Delta t} - \frac{\Delta_n[b\mathcal{W}_S](t_n)}{\Delta t} \\
&= \gamma\hbar\omega_0\left|\langle\sigma_-\rangle_{t_n}\right|^2 - \frac{\gamma\hbar\omega_0}{2} - \gamma\hbar\omega_0\left(\bar{n}_{\mathrm{th}}+\frac{1}{2}\right)\langle\sigma_z\rangle_{t_n}.
\end{aligned}
$$
(F.6)

968 • Field bipartite heat flow:

$$
\begin{aligned}
\frac{\Delta_n[b\mathcal{Q}_f](t_n)}{\Delta t} &= \frac{\Delta_n\mathcal{U}_f(t_n)}{\Delta t} - \frac{\Delta_n[b\mathcal{W}_f](t_n)}{\Delta t} \\
&= -\gamma\hbar\omega_0\left|\langle\sigma_-\rangle_{t_n}\right|^2 + \frac{\gamma\hbar\omega_0}{2} + \gamma\left(\bar{n}_{\mathrm{th}}+\frac{1}{2}\right)\left(\hbar\omega_0\langle\sigma_z\rangle_{t_n} + \langle H_D(t)\rangle_{t_n}\right) \\
&\quad + \frac{\gamma}{2}\langle H_D(t_n)\rangle_{t_n}\langle\sigma_z\rangle_{t_n}.
\end{aligned}
$$
(F.7)

969 ## F.2 Expressions of energetic quantities in terms of field averages for initial
970 coherent-thermal field

971 Here, we apply the input-output formalism presented in section 3.2 to derive measurable
972 expressions for the BQE quantities for the atom and the field. From Eq. (19) and Eq. (A.7),
973 taking $\lim_{\gamma\Delta t\to 0} t_m = t$ we can easily see that for a quasi-monochromatic field when $\omega_L \gg \Omega$,
974 i.e., within the regime of the OBEs,

$$
\begin{aligned}
\dot{\mathcal{U}}_f(t) &= \hbar\omega_L\lim_{\gamma\Delta t\to 0}\left\langle b_{x_1}^\dagger(t_m)b_{x_1}(t_m) - b_{x_0}^\dagger(t_m)b_{x_0}(t_m)\right\rangle/(\Delta t) \\
&= \hbar\omega_L\left\langle b_{\mathrm{out}}^\dagger(t)b_{\mathrm{out}}(t) - b_{\mathrm{in}}^\dagger(t)b_{\mathrm{in}}(t)\right\rangle \\
&= \hbar\omega_L(I_{\mathrm{out}} - I_{\mathrm{in}})
\end{aligned}
$$

is a good approximation for the rate of change of internal energy of the field. As we can relate the rate of change of internal energy of the atom with that of the field (see the previous section), we will now compute the $b$-work flows in terms of the field operators.

### F.2.1  Bipartite work done on the atom

Using Eq. (F.4), we have

$$\frac{\Delta_n[b\mathcal{W}_S](t_n)}{\Delta t} = \hbar\omega_0\Omega\text{Re}\left(e^{i(\omega_L-\omega_0)t_n}\langle\sigma_-\rangle_{t_n}\right) - \gamma\hbar\omega_0\left|\langle\sigma_-\rangle_{t_n}\right|^2 \tag{F.8}$$

Now we substitute $\frac{\Omega}{2}e^{-i(\omega_L-\omega_0)t_n} = \sqrt{\gamma}\langle b_{\text{in}}(t_n)\rangle_{t_n}$ (see Appendix C) and $\sqrt{\gamma}\langle\sigma_-\rangle_{t_n} = \left(\langle b_{\text{in}}(t_n)\rangle_{t_n} - \langle b_{\text{out}}(t_n)\rangle_{t_n}\right)$ (see Eq. (22)) into Eq. (F.4) and we find:

$$\begin{aligned}
\frac{\Delta_n[b\mathcal{W}_S](t_n)}{\Delta t} =&\, 2\hbar\omega_0\text{Re}\left(\langle b_{\text{in}}^\dagger(t_n)\rangle_{t_n}\left(\langle b_{\text{in}}(t_n)\rangle_{t_n} - \langle b_{\text{out}}(t_n)\rangle_{t_n}\right)\right)\\
&-\hbar\omega_0\left(\langle b_{\text{in}}^\dagger(t_n)\rangle_{t_n} - \langle b_{\text{out}}^\dagger(t_n)\rangle_{t_n}\right)\left(\langle b_{\text{in}}(t_n)\rangle_{t_n} - \langle b_{\text{out}}(t_n)\rangle_{t_n}\right)\\
=&-2\hbar\omega_0\text{Re}\left(\langle b_{\text{in}}^\dagger(t_n)\rangle_{t_n}\langle b_{\text{out}}(t_n)\rangle_{t_n}\right) + 2\hbar\omega_0\text{Re}\left(\langle b_{\text{in}}^\dagger(t_n)\rangle_{t_n}\langle b_{\text{out}}(t_n)\rangle_{t_n}\right)\\
&-\hbar\omega_0\left(\left|\langle b_{\text{in}}(t_n)\rangle_{t_n}\right|^2 + \left|\langle b_{\text{out}}(t_n)\rangle_{t_n}\right|^2\right) + 2\hbar\omega_0\left|\langle b_{\text{in}}(t_n)\rangle_{t_n}\right|^2\\
=&\,\hbar\omega_0\left(\left|\langle b_{\text{in}}(t_n)\rangle_{t_n}\right|^2 - \left|\langle b_{\text{out}}(t_n)\rangle_{t_n}\right|^2\right).
\end{aligned} \tag{F.9}$$

The above equation shows that the work done on the atom by a propagating field (Eq. (45)) can be attributed to the change of the first moment of the latter and is measurable.

### F.2.2  Bipartite work done on the propagating field

We begin with Eq. (F.5),

$$\begin{aligned}
\frac{\Delta_n[b\mathcal{W}_f](t_n)}{\Delta t} =&-\frac{\hbar\omega_L\Omega}{2}\left(e^{-i(\omega_L-\omega_0)t_n}\langle\sigma_+\rangle_{t_n} + e^{i(\omega_L-\omega_0)t_n}\langle\sigma_-\rangle_{t_n}\right)\\
&+\gamma\hbar\omega_0\left|\langle\sigma_-\rangle_{t_n}\right|^2 - \frac{\gamma}{2}\langle H_D(t_n)\rangle_{t_n}\langle\sigma_z\rangle_{t_n}.
\end{aligned} \tag{F.10}$$

As done in the previous section, we substitute $\frac{\Omega}{2}e^{-i(\omega_L-\omega_0)t_n} = \sqrt{\gamma}\langle b_{\text{in}}(t_n)\rangle_{t_n}$ (see Appendix C) and $\sqrt{\gamma}\langle\sigma_-\rangle_{t_n} = \left(\langle b_{\text{in}}(t_n)\rangle_{t_n} - \langle b_{\text{out}}(t_n)\rangle_{t_n}\right)$ (see Eq. (22)) in the first term of Eq. (F.5). Adding and subtracting $\hbar\omega_L\left|\langle b_{\text{out}}(t_n)\rangle_{t_n}\right|^2$ we find:

$$\begin{aligned}
\frac{\Delta_n[b\mathcal{W}_f](t_n)}{\Delta t} =&\,\hbar\omega_L\left(\left|\langle b_{\text{out}}(t_n)\rangle_{t_n}\right|^2 - \left|\langle b_{\text{in}}(t_n)\rangle_{t_n}\right|^2\right)\\
&+\gamma\hbar(\omega_0-\omega_L)\left|\langle\sigma_-\rangle_{t_n}\right|^2 - \frac{\gamma}{2}\langle H_D(t_n)\rangle_{t_n}\langle\sigma_z\rangle_{t_n}
\end{aligned} \tag{F.11}$$

Up to a relative error of $\mathcal{O}(\Omega/\omega_0, (\omega_L-\omega_0)/\omega_0)$,

$$b\dot{\mathcal{W}}_f(t_n) = \hbar\omega_L\left(\left|\langle b_{\text{out}}(t_n)\rangle_{t_n}\right|^2 - \left|\langle b_{\text{in}}(t_n)\rangle_{t_n}\right|^2\right)$$

which is the measurable expression of the rate of bipartite work done on the field by the atom (see Eq. (45) and Section 8). This is exact at steady state as, $\frac{\gamma}{2}\langle H_D(t_\infty)\rangle_{t_\infty}\langle\sigma_z\rangle_{t_\infty} = \gamma\hbar(\omega_0-\omega_L)\left|\langle\sigma_-\rangle_{t_\infty}\right|^2$. Comparing with Eq. (45), we find the simple relation $b\dot{\mathcal{W}}_S = -(\hbar\omega_0/\hbar\omega_L)b\dot{\mathcal{W}}_f$ which is exact at steady state.

### F.3   Expressions of energetic quantities at steady state

By definition, at steady state $\langle \dot{\sigma}_x \rangle_{t_\infty} = \langle \dot{\sigma}_y \rangle_{t_\infty} = \langle \dot{\sigma}_z \rangle_{t_\infty} = 0$, implying $\dot{\mathcal{U}}_S(t_\infty) = \dot{\mathcal{U}}_f(t_\infty) = \dot{\mathcal{V}}(t_\infty) = 0$ (see section 5). This results in the atom and the field exchanging $b$-work and $b$-heat such that $b\dot{\mathcal{W}}_{S(f)}(t_\infty) = -b\dot{\mathcal{Q}}_{S(f)}(t_\infty)$. As shown in Appendix F.2.2, at steady state, $b\dot{\mathcal{W}}_f(t_\infty) = -(\omega_L/\omega_0)b\dot{\mathcal{W}}_S(t_\infty)$. Hence, the all energetic quantities at steady state can be derived from $b\dot{\mathcal{W}}_S(t_\infty)$ and $b\dot{\mathcal{W}}_S^s(t_\infty)$ which can be computed using Eq. (F.4). The steady state averages are expressed in terms of the saturation parameter $s = 2\Omega^2/\left(4(\omega_L - \omega_0)^2 + \gamma^2(2\bar{n}_{\text{th}} + 1)^2\right)$:

$$b\dot{\mathcal{W}}_S(t_\infty) = -\frac{\gamma\hbar\omega_0}{2(2\bar{n}_{\text{th}} + 1)^2}\left(\frac{s}{(1+s)^2}\right) + \frac{\gamma\hbar\omega_0}{2}\left(\frac{s}{1+s}\right), \tag{F.12}$$

$$b\dot{\mathcal{W}}_S^s(t_\infty) = -\frac{\gamma\hbar\omega_0}{2(2\bar{n}_{\text{th}} + 1)^2}\left(\frac{s}{(1+s)^2}\right). \tag{F.13}$$

### F.4   Average coupling energy for initial coherent-thermal field and atom using the collisional model

Here, we derive the average value of the interaction picture coupling Hamiltonian $V(t)$. We start with its discrete-time version given in Eq. (13) of the main text

$$V_n = i\sqrt{\gamma\Delta t}\left(\sigma_+ b_n - b_n^\dagger \sigma_-\right). \tag{F.14}$$

Throughout we will use the fact that $\text{Tr}_f\{V_n \eta_n^\beta\} = 0$. The change of the coupling energy after the $n^{\text{th}}$ collision is

$$\frac{\Delta t}{\hbar}\Delta_n \mathcal{V} = \frac{\Delta t}{\hbar}(\mathcal{V}(t_{n+1}) - \mathcal{V}(t_n))$$

$$= \text{Tr}\left\{V_{n+1}\rho^{'}(t_{n+1})\right\} - \text{Tr}\left\{V_n \rho_S(t_n)\bigotimes_m \mathcal{D}(\alpha_m)\eta_m^\beta \mathcal{D}^\dagger(\alpha_m)\right\}, \tag{F.15}$$

where $\rho^{'}(t_{n+1})$ is the bipartite state of the atom and the field after the $n^{\text{th}}$ unit has interacted (tracing over all other $(n-1)$ units that have already interacted). The state of the field may contain all the field units that have yet to interact. As stated in Section 3.2 of the main text, the evolution of the bipartite state reads

$$\rho^{'}(t_{n+1}) = \mathcal{U}_n\left(\rho_S(t_n)\bigotimes_m \mathcal{D}(\alpha_m)\eta_m^\beta \mathcal{D}^\dagger(\alpha_m)\right)\mathcal{U}_n^\dagger. \tag{F.16}$$

Expanding the unitary to the first order in $\gamma\Delta t$,

$$\rho^{'}(t_{n+1}) = \rho_S(t_n)\bigotimes_m \mathcal{D}(\alpha_m)\eta_m^\beta \mathcal{D}^\dagger(\alpha_m) - i\left[V_n, \rho_S(t_n)\bigotimes_m \mathcal{D}(\alpha_m)\eta_m^\beta \mathcal{D}^\dagger(\alpha_m)\right]$$

$$- \frac{1}{2}\left[V_n, \left[V_n, \rho_S(t_n)\bigotimes_m \mathcal{D}(\alpha_m)\eta_m^\beta \mathcal{D}^\dagger(\alpha_m)\right]\right]. \tag{F.17}$$

Substituting this in the coupling energy, we get

$$\frac{\Delta t}{\hbar}\Delta_n \mathcal{V} = \text{Tr}\left\{(V_{n+1} - V_n)\rho_S(t_n)\bigotimes_m \mathcal{D}(\alpha_m)\eta_m^\beta \mathcal{D}^\dagger(\alpha_m)\right\}$$

$$- i\text{Tr}\left\{[V_{n+1}, V_n]\rho_S(t_n)\bigotimes_m \mathcal{D}(\alpha_m)\eta_m^\beta \mathcal{D}^\dagger(\alpha_m)\right\}$$

$$- \frac{1}{2}\text{Tr}\left\{V_{n+1}\left[V_n, \left[V_n, \rho_S(t_n)\bigotimes_m \mathcal{D}(\alpha_m)\eta_m^\beta \mathcal{D}^\dagger(\alpha_m)\right]\right]\right\}. \tag{F.18}$$

1014  We can trace over all the units that are not involved in the evolution and use the cyclic property
1015  of the trace to apply the displacement on $V_{n(n+1)}$ to get

$$\Delta_n \mathcal{V} \Delta t = \text{Tr}\left\{ \left(H_D\left(t_{n+1}\right)\Delta t + \hbar V_{n+1}\right)\left(\rho_S\left(t_n\right) \otimes \eta_{n+1}^\beta\right) - \left(H_D\left(t_n\right)\Delta t + \hbar V_n\right)\rho_S\left(t_n\right) \otimes \eta_n^\beta\right\}$$

$$-i\text{Tr}\left\{\left[H_D\left(t_{n+1}\right)\Delta t + \hbar V_{n+1}, H_D\left(t_n\right)\Delta t + \hbar V_n\right]\left(\rho_S\left(t_n\right)\bigotimes_m \eta_m^\beta\right)\right\}$$

$$-\frac{1}{2}\text{Tr}\left\{\left[\left[H_D\left(t_{n+1}\right)\Delta t + \hbar V_{n+1}, H_D\left(t_n\right)\Delta t + \hbar V_n\right], H_D\left(t_n\right)\Delta t + \hbar V_n\right]\rho_S\left(t_n\right)\bigotimes_m \eta_m^\beta\right\}.$$

$$\text{(F.19)}$$

1016  Now we use that $\text{Tr}\left\{V_{n+1}\rho_S\left(t_n\right)\eta_{n+1}^\beta\right\} = 0$ and $\Delta H_D\left(t_n\right) = H_D\left(t_{n+1}\right) - H_D\left(t_n\right)$,

$$\Delta_n \mathcal{V} \Delta t = \text{Tr}_S\left\{\Delta H_D\left(t_n\right)\Delta t \rho_S\left(t_n\right)\right\}$$

$$-i\text{Tr}\left\{\left[H_D\left(t_{n+1}\right)\Delta t + \hbar V_{n+1}, H_D\left(t_n\right)\Delta t + \hbar V_n\right]\left(\rho_S\left(t_n\right)\bigotimes_m \eta_n^\beta\right)\right\}$$

$$-\frac{1}{2}\text{Tr}\left\{\left[\left[H_D\left(t_{n+1}\right)\Delta t + \hbar V_{n+1}, H_D\left(t_n\right)\Delta t + V_n\right], H_D\left(t_n\right)\Delta t + \hbar V_n\right]\rho_S\left(t_n\right)\bigotimes_m \eta_m^\beta\right\}.$$

$$\text{(F.20)}$$

1017  Re-writing the second and third lines we get

$$\text{Tr}\left\{\left[H_D\left(t_{n+1}\right)\Delta t + \hbar V_{n+1}, H_D\left(t_n\right)\Delta t + \hbar V_n\right]\left(\rho_S\left(t_n\right)\bigotimes_m \eta_m^\beta\right)\right\}$$

$$= \text{Tr}\left\{H_D\left(t_{n+1}\right)\Delta t\left[H_D\left(t_n\right)\Delta t + \hbar V_n, \left(\rho_S\left(t_n\right)\otimes \eta_n^\beta\right)\right]\right\}$$

$$+ \text{Tr}\left\{\left[\hbar V_{n+1}, H_D\left(t_n\right)\Delta t + \hbar V_n\right]\left(\rho_S\left(t_n\right)\otimes \eta_{n+1}^\beta\right)\right\}$$

1018  where, we see that $\text{Tr}\left\{\left[\hbar V_{n+1}, H_D\left(t_n\right)\Delta t + \hbar V_n\right]\left(\rho_S\left(t_n\right)\otimes \eta_n^\beta\right)\right\} = 0$ as $V_{n+1}$ gets traced only
1019  with $\eta_{n+1}^\beta$ and hence, $\text{Tr}\left\{V_{n+1}V_n \eta_{n+1}^\beta\right\} = 0$. Defining $\tilde{V}_n = H_D\left(t_n\right)\Delta t + \hbar V_n$

$$\text{Tr}\left\{\left[\left[H_D\left(t_{n+1}\right)\Delta t + \hbar V_{n+1}, \tilde{V}_n\right], \tilde{V}_n\right]\rho_S\left(t_n\right)\bigotimes_m \eta_m^\beta\right\} = \text{Tr}\left\{\left[\left[H_D\left(t_{n+1}\right)\Delta t, \tilde{V}_n\right], \tilde{V}_n\right]\rho_S\left(t_n\right)\bigotimes_m \eta_m^\beta\right\}$$

$$+ \text{Tr}\left\{\left[\left[\hbar V_{n+1}, \tilde{V}_n\right], \tilde{V}_n\right]\rho_S\left(t_n\right)\bigotimes_m \eta_m^\beta\right\}.$$

1020  Again, the second term is zero as $V_{n+1}$ gets traced only with $\eta_{n+1}^\beta$ and hence,
1021  $\text{Tr}\left\{V_{n+1}\tilde{V}_n \eta_{n+1}^\beta\right\} = 0$. Finally, we get

$$\Delta_n \mathcal{V} \Delta t = \text{Tr}_S\left\{\Delta_n H_D\left(t_n\right)\Delta t \rho_S\left(t_n\right)\right\} + \text{Tr}\left\{H_D\left(t_{n+1}\right)\Delta t\left[H_D\left(t_n\right)\Delta t + \hbar V_n, \left(\rho_S\left(t_n\right)\otimes \eta_n^\beta\right)\right]\right\}$$

$$+ \text{Tr}\left\{H_D\left(t_{n+1}\right)\Delta t\left[\left(H_D\left(t_n\right)\Delta t + \hbar V_n\right), \left[\left(H_D\left(t_n\right)\Delta t + \hbar V_n\right), \rho_S\left(t_n\right)\bigotimes_m \eta_m^\beta\right]\right]\right\}$$

$$= \text{Tr}_S\left\{\Delta_n H_D\left(t_n\right)\Delta t \rho_S\left(t_n\right)\right\}$$

$$+ \text{Tr}_S\left\{H_D\left(t_{n+1}\right)\Delta t \text{Tr}_f\left\{\tilde{\mathcal{U}}_n\left(\rho_S\left(t_n\right)\bigotimes_m \eta_m^\beta\right)\tilde{\mathcal{U}}_n^\dagger - \rho_S\left(t_n\right)\bigotimes_m \eta_m^\beta\right\}\right\}. \quad \text{(F.21)}$$

where $\tilde{\mathcal{U}}_n = \mathcal{D}^\dagger(\alpha_n)\mathcal{U}_n\mathcal{D}(\alpha_n)$. Noticing that

$$\text{Tr}_f\left\{\tilde{\mathcal{U}}_n\rho_S\left(t_n\right)\bigotimes_m \eta_m^\beta \tilde{\mathcal{U}}_n^\dagger - \rho_S\left(t_n\right)\bigotimes_m \eta_m^\beta\right\} = \rho_S(t_{n+1}) - \rho_S(t_n) = \Delta_n \rho_S$$

is the change of the atom's state after the $n^{\text{th}}$ collision, we get $\Delta_n \mathcal{V} = \Delta_n \langle H_D(t_n) \rangle_{t_n}$, which in the continuous limit results in,

$$\dot{\mathcal{V}}(t) = \frac{d}{dt} \langle H_D(t) \rangle_t \implies \Delta \mathcal{V}(t) = \Delta \langle H_D(t) \rangle_t. \tag{F.22}$$

## G Energetic relations between Open and Autonomous Scenarios

Here we compare the energetic framework applied in the standard open approach with the one detailed in Section 5 for the autonomous scenario. We first relate $b$-work done on the field with the standard definition of work $\dot{W}$. We begin showing that out of resonance, at any time $t$, $b\dot{\mathcal{W}}_f = -\dot{W} - \text{Tr}_S\{\rho_S^{\text{(lab)}}(t_n)\Delta_n \mathcal{H}_S^{s\text{(lab)}}(t_n)/\Delta t\}$, where for simplicity we define $\dot{W}_{\text{self}}(t) = \text{Tr}_S\{\rho_S^{\text{(lab)}}(t_n)\Delta_n \mathcal{H}_S^{s\text{(lab)}}(t_n)/\Delta t\}$. We can then use this result to get Eq. (48) and (49) of the main text by taking $\mathcal{V} = \mathcal{V}^{\otimes} = \mathcal{V}^{\chi} = 0$ which is true at resonance [29] and also show Eq. (47) out of resonance. Here, we work in the lab frame with respect to the atom (still rotating with $H_f$) and remove the notation $^{\text{lab}}$ for the operators in this section. Substituting $\Delta_n \rho^{\otimes}$ (see Eq. (32)) in Eq. (41) for the field, we find:

$$\begin{aligned}
\Delta_n \left[ b\mathcal{W}_f \right] &= \text{Tr}\left\{ H_f \Delta_n \rho^{\otimes} \right\} \\
&= \text{Tr}_f \left\{ H_f \Delta_n \rho_f^{(1)} \right\} + \text{Tr}_f \left\{ H_f \Delta_n \rho^{(2,f)} \right\} + \text{Tr}_f \left\{ H_f \Delta_n \rho^{(2,S)} \right\} \\
&= \text{Tr}_f \left\{ H_f \Delta_n \rho_f^{(1)} \right\} + \text{Tr}_f \left\{ H_f \Delta_n \rho_f^{(2,f)} \right\},
\end{aligned} \tag{G.1}$$

where, the contribution of $\Delta_n \rho_f^{(2,S)}$ is negligible (see Appendix D). The first and second order terms of the $b$-work done on the field are equal and opposite to the work and self-work respectively. Substituting Eq. (D.1) in the first order contribution, it becomes:

$$\text{Tr}\left\{ H_f \Delta_n \rho_f^{(1)} \right\} = -i\text{Tr}\left\{ \left[ H_f, V_n \right] \rho_n \right\}. \tag{G.2}$$

Now, we use that in the interaction picture, the operators evolve as $\left[ H_f, V_n \right] = -i\hbar \frac{\Delta_n V_n}{\Delta t}$. Substituting this in the above, we find

$$\begin{aligned}
\text{Tr}\left\{ H_f \Delta_n \rho_f^{(1)} \right\} &= -i\text{Tr}\left\{ \left[ H_f, V_n \right] \rho_n \right\} \\
&= -\hbar \text{Tr}\left\{ \frac{\Delta_n V_n}{\Delta t} \rho_n \right\} \\
&= -\text{Tr}\left\{ \Delta_n H_D \rho_S(t_n) \right\} - \hbar \text{Tr}\left\{ \frac{\Delta_n V_n}{\Delta t} \rho_S \otimes \eta_n^{\beta} \right\} \\
&= -\text{Tr}\left\{ \Delta_n H_D \rho_S(t_n) \right\} = -\Delta_n W.
\end{aligned} \tag{G.3}$$

Hence, the first order contribution to the field's $b$-work is equal and opposite to the work done on the atom $\Delta_n W$. This can be seen as a result of $\Delta_n \rho_f^{(1)}$ arising from stimulated emission processes of the atom and hence, leads to an energetic contribution which is equal and opposite to the standard work done on the atom $-\dot{W}$.

Similarly, substituting Eq. (30) after taking the trace over the atom, in the second order

contribution, we find:

$$
\begin{aligned}
\text{Tr}\left\{H_f \Delta_n \rho_f^{(2,f)}\right\} &= -\frac{1}{2}\text{Tr}\left\{\left[H_f, V_n\right]\left[\langle V_n\rangle_S, \rho(t_n)\right]\right\} \\
&= \frac{i\hbar}{2}\text{Tr}\left\{\left[\frac{\Delta_n V_n}{\Delta t}, \langle V_n\rangle_S\right]\rho(t_n)\right\} \\
&= \frac{i\hbar}{2}\text{Tr}\left\{\frac{\Delta_n}{\Delta t}\left(\left[V_n, \langle V_n\rangle_S\right]\right)\rho(t_n)\right\} - \frac{i\hbar}{2}\text{Tr}\left\{\left(\left[V_n, \frac{\Delta_n\langle V_n\rangle_S}{\Delta t}\right]\right)\rho(t_n)\right\} \\
&= -\text{Tr}\left\{\rho_S(t_n)\Delta_n\left(\mathcal{H}_S^s(t_n)\right)\right\} - \frac{i\hbar}{2}\text{Tr}\left\{\left(\left[\langle V_n\rangle_S, \frac{\Delta_n\langle V_n\rangle_S}{\Delta t}\right]\right)\rho_f(t_n)\right\},
\end{aligned}
$$

(G.4)

where, we used that $\left[V_n, \langle V_n\rangle_S\right] = \frac{2i\Delta t}{\hbar}\mathcal{H}_S^s(t_n)$. Finally it is easy to see that:

$$
-\frac{i\hbar}{2}\text{Tr}\left\{\left[\langle V_n\rangle_S, \frac{\Delta_n\langle V_n\rangle_S}{\Delta t}\right]\rho_f(t_n)\right\} = -\frac{i\hbar}{2}\text{Tr}\left\{\left[\langle V_n\rangle_S, \frac{\langle V_{n+1}\rangle_S - \langle V_n\rangle_S}{\Delta t}\right]\rho_f(t_n)\right\} = 0, \quad \text{(G.5)}
$$

and hence,

$$
\text{Tr}\left\{H_f \Delta_n \rho_f^{(2,f)}\right\} = -\text{Tr}\left\{\rho_S(t_n)\Delta_n\left(\mathcal{H}_S^s(t_n)\right)\right\} = -\Delta_n W_{\text{self}}. \quad \text{(G.6)}
$$

The $b$-work rate is finally written as:

$$
b\dot{\mathcal{W}}_f = -\dot{W} - \dot{W}_{\text{self}} \quad \text{(G.7)}
$$

where the first term is the work rate (Eq. (7) computed in the lab frame) and the we will now show that $\dot{W}_{\text{self}} = (\omega_L/\omega_0)b\dot{\mathcal{W}}_S^s$. Explicitly computing $\dot{W}_{\text{self}}$ by differentiating self-drive in Eq. (34) we find:

$$
\dot{W}_{\text{self}} = \frac{\gamma}{2}\langle H_D(t_n)\rangle_{t_n}\langle\sigma_z\rangle_{t_n} - \gamma\hbar\omega_0\left|\langle\sigma_-\rangle_{t_n}\right|^2. \quad \text{(G.8)}
$$

At resonance, $\mathcal{V}(t) = \Delta\langle H_D(t)\rangle = 0$ and hence, as $\langle H_D(t)\rangle = 0$, $\dot{W}_{\text{self}} = b\dot{\mathcal{W}}_S^s$. Using $b\dot{\mathcal{W}}_f = -b\dot{\mathcal{W}}_S$ and global energy conservation, we also find Eq. (48) and (49) of the main text.

Out of resonance, at steady state we find

$$
\frac{\gamma}{2}\langle H_D(t_\infty)\rangle_{t_\infty}\langle\sigma_z\rangle_{t_\infty} = \gamma\hbar(\omega_0 - \omega_L)\left|\langle\sigma_-\rangle_{t_\infty}\right|^2,
$$

and hence,

$$
\dot{W}_{\text{self}} = \frac{\omega_L}{\omega_0}b\dot{\mathcal{W}}_S^s. \quad \text{(G.9)}
$$

The above equation is true at all times up to a relative error of $\mathcal{O}(\Omega/\omega_0, (\omega_L - \omega_0)/\omega_0)$. Hence, we see that the second order contribution $\Delta_n\rho_f^{(2,f)}$ which results from the unitary part of the spontaneous emission process, has an energetic contribution which is equal and opposite to the self-work in the steady state regime done at the frequency of the field. With both these results, we find Eq. (47).

# H  Derivation and Validity of Second Law from the Collisional model

Here we derive Eq. (53) and prove that it provides a tighter bound over the second law given by Eq. (9) and the standard Clausius inequality.

## H.1 Derivation of the Clausius inequality

In order to derive the Clausius relation with the $b$-heat, we employ the collision model description of the interaction between the atom and the waveguide field as described in section 3.2. Here we work in the displaced frame with respect to the initial displacement of the field and denote by $\eta_n$ the state of the $n^{\text{th}}$ unit of the field in the input cell before the collision and $\eta'_n$ as the state of the same unit in the output cell after the collision in this frame. We begin the derivation with Eq. (52), where we rewrite the displacement operator (see Eq. (51)) in terms of displacements on the collisional units using Eq. (A.5)

$$
\mathcal{D}(t) = \bigotimes_{n=0}^{N} \exp\left\{ \sqrt{\gamma \Delta t} \left( \langle \sigma_+ \rangle_{t_n} b_n - \langle \sigma_- \rangle_{t_n} b_n^\dagger \right) \right\}
$$
$$
= \bigotimes_{n=0}^{N} \mathcal{D}(\varphi_n)
\tag{H.1}
$$

where, $\varphi_n = -\sqrt{\gamma \Delta t} \langle \sigma_- \rangle_{t_n}$ and $N = (t - t_0)/\Delta t$ such that, as $\gamma \Delta t \to 0$, $N \to \infty$. Substituting in Eq. (52) we get

$$
b\Sigma = -\text{Tr}\left\{ \rho_S(t) \ln \rho_S(t) \right\} + \text{Tr}\left\{ \rho(t) \ln \rho(t) \right\}
$$
$$
- \lim_{\gamma \Delta t \to 0} \sum_{n}^{N} \text{Tr}\left\{ \eta'_n \ln\left( \mathcal{D}(\varphi_n) \eta_n \mathcal{D}^\dagger(\varphi_n) \right) \right\}.
\tag{H.2}
$$

Now we use the fact that the state of the collisional unit is a thermal white noise in the displaced frame where $\eta_n = \eta_n^\beta$ to get

$$
b\Sigma = -\text{Tr}\left\{ \rho_S(t) \ln \rho_S(t) \right\} + \text{Tr}\left\{ \rho(t) \ln \rho(t) \right\}
$$
$$
+ \beta \lim_{\gamma \Delta t \to 0} \sum_{n}^{N} \text{Tr}\left\{ \mathcal{D}^\dagger(\varphi_n) \eta'_n \mathcal{D}(\varphi_n) H_f \right\} + \ln(Z_n)
$$
$$
= \beta \lim_{\gamma \Delta t \to 0} \sum_{n}^{N} \text{Tr}\left\{ \left( \mathcal{D}^\dagger(\varphi_n) \eta'_n \mathcal{D}(\varphi_n) - \eta_n \right) H_f \right\} + \Delta S_S,
\tag{H.3}
$$

where $\Delta S_S$ is the change of the entropy of the atom and we used that fact that $\text{Tr}\left\{ \rho(t) \ln \rho(t) \right\} = \text{Tr}\left\{ \rho(0) \ln \rho(0) \right\}$ as it is a closed system. Hence, the rate of change of $b\Sigma$ is given by,

$$
b\dot{\Sigma} = \frac{1}{\Delta t} \left( \Delta_n S_S + \beta \text{Tr}\left\{ \left( \mathcal{D}^\dagger(\varphi_n) \eta'_n \mathcal{D}(\varphi_n) - \eta_n \right) H_f \right\} \right),
\tag{H.4}
$$

where, $\Delta_n S_S$ is the change of the entropy of the atom during the collision with the $n^{\text{th}}$ unit of the field. Following the same strategy as the Dyson expansion to derive Eq. (18), up to an order of $\gamma \Delta t$, the action of the displacement on $\eta'_n$ is

$$
\mathcal{D}^\dagger(\varphi_n) \eta'_n \mathcal{D}(\varphi_n) = \eta'_n + i\Delta t \left[ \mathcal{H}_f(t_n), \eta_n \right] - \frac{\Delta t^2}{2} \left[ \mathcal{H}_f(t_n), \left[ \mathcal{H}_f(t_n), \eta_n \right] \right]
$$
$$
= \eta'_n + \text{Tr}_{m \neq n}\{ \Delta_n \rho_f^{(1)} + \Delta_n \rho_f^{(2,f)} \},
\tag{H.5}
$$

where, in the last line we have substituted Eq. (D.1) and Eq. (D.4) and the trace is over all collision modes expect the $n^{\text{th}}$ mode. Substituting Eq. (H.5) in Eq.(H.4), we get

$$
b\dot{\Sigma} \Delta t = \Delta_n S_S + \beta \text{Tr}\left\{ \left( \eta'_n - \eta_n \right) H_f \right\}
$$
$$
+ i\beta \text{Tr}\left\{ H_f \left( \Delta_n \rho_f^{(1)} + \Delta_n \rho_f^{(2,f)} \right) \right\}.
\tag{H.6}
$$

We identify $b\dot{\mathcal{W}}_f(t_n)\Delta t = -\frac{i}{\hbar}\mathrm{Tr}\left\{H_f\left(\Delta_n\rho_f^{(1)} + \Delta_n\rho_f^{(2,f)}\right)\right\}$ as the rate of $b$-work done on each field unit. Hence, substituting $\Delta_n\mathcal{U}_f = \mathrm{Tr}\{(\eta_n' - \eta_n)H_f\}$ and $b\dot{\mathcal{W}}_f(t_n)$ in Eq. (H.6) and taking the continuous limit we get

$$
\begin{aligned}
b\dot{\Sigma} &= \dot{S}_S + \beta \lim_{\gamma\Delta t\to 0}\frac{\Delta_n\mathcal{U}_f}{\Delta t} - \beta \lim_{\gamma\Delta t\to 0} b\dot{\mathcal{W}}_f(t_n) \\
&= \dot{S}_S + \beta\dot{\mathcal{U}}_f(t) - \beta b\dot{\mathcal{W}}_f(t) \\
&= \dot{S}_S + \beta b\dot{\mathcal{Q}}_f(t).
\end{aligned}
\tag{H.7}
$$

We also find the integrated expression:

$$
\begin{aligned}
b\Sigma &= \Delta S_S + \beta \lim_{\gamma\Delta t\to 0}\sum_n^N \Delta_n\mathcal{U}_f - \beta \lim_{\gamma\Delta t\to 0}\sum_n^N \Delta t\, b\dot{\mathcal{W}}_f(t_n) \\
&= \Delta S_S + \beta\Delta\mathcal{U}_f(t) - \beta b\mathcal{W}_f(t) \\
&= \Delta S_S + \beta b\mathcal{Q}_f(t).
\end{aligned}
\tag{H.8}
$$

At resonance, $b\mathcal{Q}_f(t) = -b\mathcal{Q}_S(t)$, using which we find the Clausius relation from Eq. (53). The positivity of $b\dot{\Sigma}$ and $b\Sigma$ arises from the positivity of the relative entropy, noting that this relative entropy before the first collision (before interaction has begun) is zero.

## H.2   Sign of $\dot{W}_{\text{self}}$

We begin with $\dot{W}_{\text{self}} = -b\dot{\mathcal{W}}_f - \dot{W} = \mathrm{Tr}\{\rho_S^{(\text{lab})}(t)d\mathcal{H}_S^{s(\text{lab})}(t)/dt\}$ (see Appendix G). Using global energy conservation this can be expressed as, $\dot{Q}_S + b\dot{\mathcal{Q}}_f = \dot{W}_{\text{self}} - \dot{\mathcal{V}}$. We first show that $\dot{W}_{\text{self}} \leq 0$. We compute

$$
\dot{W}_{\text{self}} = -i\hbar\frac{\gamma}{2}\left(\langle\dot{\sigma}_-\rangle\langle\sigma_+\rangle - \langle\dot{\sigma}_+\rangle\langle\sigma_-\rangle\right).
\tag{H.9}
$$

Using the atom's master equation, we find

$$
\langle\dot{\sigma}_-\rangle = -i\omega_0\langle\sigma_-\rangle - \frac{\Omega}{2}e^{-i\omega_L t}\langle\sigma_z\rangle - \gamma\left(\frac{2\bar{n}_{\text{th}}+1}{2}\right)\langle\sigma_-\rangle.
\tag{H.10}
$$

Introducing for readability the Bloch coordinates $q(t) = \langle\sigma_q\rangle$ for $q = x, y, z$, we obtain

$$
\dot{W}_{\text{self}} = -\frac{\gamma\hbar\omega_0}{4}\left(\left(x^2(t) + y^2(t)\right) + \frac{\Omega}{\omega_0}y(t)z(t)\right).
\tag{H.11}
$$

When sweeping the admissible atom states (i.e., verifying $x^2(t) + y^2(t) + z^2(t) \leq 1$ and $x(t), y(t), z(t) \in [-1,1]$), it is easy to show that $\dot{W}_{\text{self}} \leq 0$ except when we have both $x(t) \in [-\sqrt{-y^2(t) - \epsilon y(t)z(t)}, \sqrt{-y^2(t) - \epsilon y(t)z(t)}]$, and $y(t)(y(t) + \epsilon z(t)) < 0$, with $\epsilon = \Omega/\omega_0 \ll 1$. In this range of parameter, it is easy to check that the maximum value taken by $\dot{W}_{\text{self}}(t)$ is $\frac{\gamma\hbar\omega_0}{8}(\sqrt{1+\epsilon^2} - 1) \simeq \gamma\hbar\omega_0\frac{\Omega^2}{16\omega_0^2} \ll \gamma\hbar\omega_0$. Both the maximum positive value of $\dot{W}_{\text{self}}$ and the range of parameters allowing to have $\dot{W}_{\text{self}} \geq 0$ vanish as $\mathcal{O}(\Omega^2/\omega_0^2)$. Therefore, except in that narrow interval, where the work-like quantities are sensibly equivalent, $\dot{W}_{\text{self}} < 0$. To show that, $b\Sigma \leq \Sigma$, we need that $\dot{Q}_S + b\dot{\mathcal{Q}}_f = \dot{W}_{\text{self}} - \dot{\mathcal{V}} \leq 0$ where,

$$
\dot{W}_{\text{self}} - \dot{\mathcal{V}} = -\frac{\gamma\hbar\omega_0}{4}\left(\left(x^2(t) + y^2(t)\right) + \epsilon y(t)z(t) + 2\epsilon\left(\bar{n}_{\text{th}} + \frac{1}{2}\right)y(t) + \frac{2\epsilon(\omega_L - \omega_0))}{\gamma}x(t)\right).
\tag{H.12}
$$

Similarly, we can notice, for a quasi-resonant field, where $\epsilon\sqrt{\left(\bar{n}_{\text{th}} + \frac{1}{2}\right)^2 + (\frac{\omega_L - \omega_0}{\gamma})^2} \ll 1$, for the range of values considered, this quantity is negative, and hence, the field's bipartite heat is smaller than the heat received by the bath according to the OBEs. This finally implies a smaller entropy production for the closed analysis, i.e, $b\Sigma \leq \Sigma$.

## I    Spectral signatures of correlations

Here we derive the equations present in Section 7 starting from the output photon-number spectrum. We first relate this to the Fourier transform of the correlation function of field at position $x_1$ and then find the contributions of the different processes during the interaction in Eq. (32) to the spectrum. We do this in order to derive Eq. (61) and the form of $\dot{S}^\chi(\omega)$ to relate it to the inelastic component of the Mollow triplet.

### I.1    Spectrum in terms of collision modes

In this section, we relate the Fourier transform of the correlation function of field at position $x_1$ to the output photon-number spectrum $\mathrm{Tr}\{a_k^\dagger a_k \rho_f(t_\infty)\}$. The state $\rho_f(t_\infty)$ is the state of the field at long times, it includes both, a large number of units that have already interacted and the ones that will interact even after the atom has already reached its limit cycle (or steady state ($\rho_{ss}$) if the atom is in the frame rotating with the laser frequency $\omega_L$). This field state is given by

$$\rho_f(t_\infty) = \mathrm{Tr}_S\left\{ \Pi_{k=0}^\infty U_k(\rho_S(t_0) \otimes \rho_{\mathrm{in}}) \Pi_{k=0}^\infty U_k^\dagger \right\}. \tag{I.1}$$

We use this state because the measurement of a single frequency requires access to a large number of temporal correlations created between all the *interacted units*. We hence, start with the Fourier transform of the field correlation function at position $x_1$ at a frequency $\omega_i$ and at some time $t_j$. Using Eq. (A.5) we find:

$$
\frac{\Delta t}{2\pi} \sum_{m,n} e^{-i\Delta\omega_i t_m} e^{i\Delta\omega_i t_n} \mathrm{Tr}\left\{ b_{x_1}^\dagger(t_m) b_{x_1}(t_n) \rho_f(t_j) \right\}
$$

$$
= \frac{\Delta t^2}{2\pi\gamma} \sum_{m,n} \sum_{k,l} g_k g_l^* e^{-i(\omega_i - \omega_l)t_m} e^{i(\omega_i - \omega_k)t_n} \mathrm{Tr}\left\{ a_l^\dagger a_k \rho_f(t_j) \right\}
$$

$$
\times e^{-i(\omega_l - \omega_k)\frac{\Delta t}{2}} \mathrm{sinc}\left( \frac{1}{2}\Delta\omega_k \Delta t \right) \mathrm{sinc}\left( \frac{1}{2}\Delta\omega_i \Delta t \right)
$$

$$
= \frac{L\Delta t}{2\pi v\gamma} \sum_n \sum_k g_k g_i^* e^{-i(\omega_i - \omega_k)t_n} \mathrm{Tr}\left\{ a_i^\dagger a_k \rho_f(t_j) \right\}
$$

$$
\times e^{-i(\omega_i - \omega_k)\frac{\Delta t}{2}} \mathrm{sinc}\left( \frac{1}{2}\Delta\omega_k \Delta t \right) \mathrm{sinc}\left( \frac{1}{2}\Delta\omega_i \Delta t \right)
$$

$$
= \mathrm{Tr}\left\{ \sum_k \delta_D(\omega_k - \omega_i) a_k^\dagger a_k \rho_f(t_j) \right\} \mathrm{sinc}^2\left( \frac{1}{2}\Delta\omega_i \Delta t \right) \frac{\sum_k g_k^2 \delta_D(\omega_k - \omega_i)}{\gamma}
$$

$$
\simeq \; S(\omega_i, t_j), \tag{I.2}
$$

where $\Delta\omega_i = \omega_i - \omega_0$, $L$ is the typical length of the waveguide and $v$ is the group velocity of the light. $L/v$ is required to get the density of modes used in defining a Kronecker delta. The approximation in the last line holds for $\omega_i \in [\omega_0 - \pi/2\Delta t, \omega_0 + \pi/2\Delta t]$, that is, the range of frequencies spanned by the collision modes $b_n$. On this frequency interval, the sinc function can be approximated by 1 and $\sum_k g_k^2 \delta_D(\omega_k - \omega_i) \simeq \gamma$, as ensured by the inequality $\Delta t \gg \tau_c \geq \Delta_{\mathrm{bw}}^{-1}$. As $\Omega, (\omega_L - \omega_0) \ll \Delta t^{-1}$ in the regime of the OBEs, the Mollow triplet (fluorescence spectrum) is fully contained in this range of frequencies. Note that the field spectrum outside this frequency range could be in principle computed from the modes $b_{n,j>0}$, evolving quasi-freely, see Appendix A.

    We      eventually      deduce      the      expression      of      spectral      flow,      verifying

1137    $\Delta t \sum_n \dot{S}\left(\omega_i, t_j; t_n\right) = S(\omega_i, t_j)$, as a function of the $b_{x_1}$ mode:

$$\dot{S}\left(\omega_i, t_j; t_n\right) = \frac{1}{\pi} \mathrm{Re} \sum_p e^{-i\Delta\omega_i t_{n+p}} e^{i\Delta\omega_i t_n} \mathrm{Tr}\left\{ b_{x_1}^\dagger\left(t_{n+p}\right) b_{x_1}\left(t_n\right) \rho_f\left(t_j\right) \right\}, \qquad \text{(I.3)}$$

1138    with the input (output) spectral flow being obtained by setting $t_j = t_0 = 0$ ($t_j = t_\infty \gg \gamma^{-1}$).
1139    We re-write the output spectrum in terms of the collision modes, introducing the delay time
1140    $(t_p = t_{n+p} - t_n)$:

$$\dot{S}_{\mathrm{out}}\left(\omega_i, t_n\right) = \frac{1}{\pi} \mathrm{Re} \sum_p e^{-i\Delta\omega_i t_p} \mathrm{Tr}_f \{ b_{n+p}^\dagger b_n \rho_f(t_\infty) \}, \qquad \text{(I.4)}$$

1141    which in the continuous limit gives:

$$\dot{S}_{\mathrm{out}}(\omega, t) = \frac{1}{\pi} \mathrm{Re} \int_0^{t_\infty} d\tau \, e^{-i(\omega-\omega_0)\tau} \left\langle b_{\mathrm{out}}^\dagger\left(t + \tau\right) b_{\mathrm{out}}\left(t\right) \right\rangle_{t_\infty}. \qquad \text{(I.5)}$$

1142    Note that above the expressions in terms of the delay times is equivalent to Eq. (I.2) as the
1143    real part is taken in the expression.

## I.2   Separating signatures of correlations

1145    Here, we will separate the contribution to the spectrum that appear due to the evolution of
1146    correlations during the interaction and in doing so derive Eq. (61) and the form of $\dot{S}^\chi(\omega)$. We
1147    are interested in its effect on the Mollow triplet and hence, we will only consider collisions
1148    after the atom has reached its limit cycle, that is $t_n \gg \gamma^{-1}$ or equivalently, that $n$ is larger than
1149    a number of collisions $n_{ss}$ required to reach the limit cycle. Singling out the input spectral
1150    flow, we obtain:

$$\dot{S}_{\mathrm{out}}\left(\omega_i, t_n\right) = \dot{S}_{\mathrm{in}}\left(\omega_i, t_n\right) + \frac{1}{\pi} \mathrm{Re} \sum_p e^{-i(\omega_i-\omega_0)t_p} \mathrm{Tr}_f \left\{ b_{n+p}^\dagger b_n \Delta_{n+p,n} \rho_f \right\}, \qquad \text{(I.6)}$$

1151      where,

$$\Delta_{n+p,n} \rho_f = \mathrm{Tr}_S \left\{ U_{n+p} \eta_{n+p} \mathcal{E}_S^{t_{n+p}-t_n} \left[ U_n \eta_n \rho_S\left(t_n\right) U_n^\dagger \right] U_{n+p}^\dagger \right\}, \qquad \text{(I.7)}$$

1152    and $\mathcal{E}_S^{t_{n+p}-t_n}$ is the evolution super-operator for the atom state over time $t_p$ and is given by

$$\mathcal{E}_S^{t_{n+p}-t_n}\left[\rho_S\right] = \prod_{k=n}^{n+p} \mathrm{Tr}_f \left\{ U_n \rho_S \eta_n U_n^\dagger \right\}. \qquad \text{(I.8)}$$

1153    For simplicity, we define for $n \geq n_{ss}$,

$$G_{np} = \mathrm{Tr}_f \left\{ b_{n+p}^\dagger b_n \Delta_{n+p,n} \rho_f \right\}, \qquad \text{(I.9)}$$

1154    such that $\dot{S}_{\mathrm{out}}\left(\omega_i\right) = \dot{S}_{\mathrm{in}}\left(\omega_i\right) + (1/2\pi) \sum_p e^{-i(\omega_i-\omega_0)t_p} G_{np}$. The term $G_{np}$ contains the
1155    change of two collisional units of the field. To separate the correlation induced evolution
1156    during these two collisions, we first separate that for the first collision and hence find
1157    $\Delta_{n+p,n} \rho_f = \Delta_{n+p,n} \rho_f^\otimes + \Delta_{n+p,n} \rho_f^\chi$, where

$$\Delta_{n+p,n} \rho_f^\otimes = \mathrm{Tr}_S \left\{ U_{n+p} \eta_{n+p} \mathcal{E}_S^{t_{n+p}-t_n} \left[ \Delta_n \rho^\otimes + \rho_S\left(t_n\right) \otimes \eta_n \right] U_{n+p}^\dagger \right\} - \eta_{n+p} \otimes \eta_n, \qquad \text{(I.10)}$$

$$\Delta_{n+p,n} \rho_f^\chi = \mathrm{Tr}_S \left\{ U_{n+p} \eta_{n+p} \mathcal{E}_S^{t_{n+p}-t_n} \left[ \Delta_n \rho^\chi \right] U_{n+p}^\dagger \right\}. \qquad \text{(I.11)}$$

1158  Their contribution to the spectrum is separated as $G_{np} = G_{np}^{\otimes} + G_{np}^{\chi}$ with

$$G_{np}^{\otimes(\chi)} = \text{Tr}_f \left\{ b_{n+p}^{\dagger} b_n \Delta_{n+p,n} \rho_f^{\otimes(\chi)} \right\}. \tag{I.12}$$

1159  We will show that this separation is enough to separate the effect of the correlation induced
1160  evolution, as the evolution induced by the correlation in the second collision has negligible
1161  contribution to $G_{np}^{\otimes}$. We will now compute the two contributions to $G_{np}$. Substituting Eq.(32)
1162  in the above we find:

$$\begin{aligned}
G_{np}^{\otimes} = &\text{Tr}_{S,n+p,n} \left\{ b_{n+p}^{\dagger} b_n U_{n+p} \eta_{n+p} \left[ \rho_S\left(t_{n+p}\right) \otimes \Delta_n \rho_f^{(1)} + \rho_S\left(t_{n+p}\right) \otimes \eta_n \right] U_{n+p}^{\dagger} \right\} \\
&+ \text{Tr}_{S,n+p,n} \left\{ b_{n+p}^{\dagger} b_n U_{n+p} \eta_{n+p} \mathcal{E}_S^{t_{n+p}-t_n} \left[ \Delta_n \rho^{(2,S)} + \Delta_n \rho^{(2,f)} \right] U_{n+p}^{\dagger} \right\} - \left\langle b_{n+p}^{\dagger} \right\rangle_{t_{n+p}} \langle b_n \rangle_{t_n},
\end{aligned} \tag{I.13}$$

1163  and

$$G_{np}^{\chi} = \text{Tr}_{S,n+p,n} \left\{ b_{n+p}^{\dagger} b_n U_{n+p} \eta_{n+p} \mathcal{E}_S^{t_{n+p}-t_n} \left[ \Delta_n \chi^{(1)} + \Delta_n \rho^{(2,\chi)} \right] U_{n+p}^{\dagger} \right\}, \tag{I.14}$$

1164  where, we used that the contribution of $\Delta_n \rho_S^{(1)}$ is of order $(\omega_L - \omega_0) \mathcal{O}\left( \Omega \Delta t \sqrt{\frac{\Delta t}{\gamma}} \right)$ as $\rho_S(t_n)$
1165  is in its limit cycle and $\Delta_n \rho_S^{(2,S)} = 0$.

1166

1167  Let us first compute $G_{np}^{\otimes}$. Substituting Eq. (30) in $G_{np}^{\otimes}$, we find that the contribution of
1168  $\Delta_n \rho^{(2,S)}$ is of order $\mathcal{O}\left( \Omega \Delta t \sqrt{\gamma \Delta t} \right)$ while that of $\Delta_n \rho^{(2,f)}$ is of order $(1 + \bar{n}_{\text{th}}) o(\gamma \Delta t)$. Using
1169  Eq. (D.1), we find $\text{Tr}\left\{ b_n \Delta_n \rho_f^{(1)} \right\} = -\sqrt{\gamma \Delta t} \langle \sigma_- \rangle_{t_n}$ and hence,

$$\begin{aligned}
G_{np}^{\otimes} = &e^{-i(\omega_L - \omega_0)t_p} \text{Tr}_{S,n+p} \left\{ b_{n+p}^{\dagger} \left[ -\sqrt{\gamma \Delta t} \langle \sigma_- \rangle_{t_n} + \langle b_n \rangle_{t_n} \right] U_{n+p} \left( \rho_S\left(t_{n+p}\right) \otimes \eta_{n+p} \right) U_{n+p}^{\dagger} \right\} \\
&- \left\langle b_{n+p}^{\dagger} \right\rangle_{t_{n+p}} \langle b_n \rangle_{t_n}.
\end{aligned}$$

1170  The action of the second collision, similarly gives

$$G_{np}^{\otimes} = -\sqrt{\gamma \Delta t} \left( \langle \sigma_- \rangle_{t_n} \left\langle b_{n+p}^{\dagger} \right\rangle_{t_{n+p}} + \langle b_n \rangle_{t_n} \langle \sigma_+ \rangle_{t_{n+p}} \right) + \gamma \Delta t \langle \sigma_+ \rangle_{t_{n+p}} \langle \sigma_- \rangle_{t_n}. \tag{I.15}$$

1171  where, we use that $\text{Tr}_{S,n+p} \left\{ b_{n+p}^{\dagger} \Delta_{n+p} \chi^{(1)} \right\} = 0$, $\text{Tr}_{S,n+p} \left\{ \Delta_{n+p} \rho_S^{(1)} \otimes b_{n+p}^{\dagger} \rho_f\left(t_{n+p}\right) \right\} = 0$,
1172  $\text{Tr}_{S,n+p} \left\{ \rho_S\left(t_{n+p}\right) \otimes b_{n+p}^{\dagger} \Delta_{n+p} \rho_f^{(1)} \right\} \quad = \quad -\sqrt{\gamma \Delta t} \langle \sigma_+ \rangle_{t_{n+p}}$ and
1173  $\text{Tr}_{S,n+p} \left\{ \rho_S(t_{n+p}) \otimes b_{n+p}^{\dagger} \eta_{n+p} \right\} = \left\langle b_{n+p}^{\dagger} \right\rangle_{t_{n+p}}$. As one can notice, there is no contribution
1174  from the correlation induced evolution during the second collision to $G_{np}^{\otimes}$. Hence, Eq.(I.14)
1175  is enough to separate the contributions of the correlations on the spectrum.

1176

1177  Now let us compute $G_{np}^{\chi}$. Using Eq. (30), we find that the contribution of the second order
1178  correlation evolution is $\text{Tr}_n \left\{ b_n \Delta_n \rho^{(2,\chi)} \right\} = o(\gamma \Delta t)$. Therefore, the only contribution from the
1179  first collision is from the creation of correlation during dynamics at first order. Using Eq. (29)
1180  in $G_{np}^{\chi}$, and taking the trace over the first collisional unit (mode $n$) we find

$$\begin{aligned}
G_{np}^{\chi} = &-\sqrt{\gamma \Delta t} \text{Tr}_{S,n+p} \left\{ b_{n+p}^{\dagger} U_{n+p} \left( \mathcal{E}_S^{t_{n+p}-t_n} \{\sigma_- \rho_S(t_n)\} \otimes \eta_{n+p} \right) U_{n+p}^{\dagger} \right\} \\
&+ \bar{n}_{\text{th}} \sqrt{\gamma \Delta t} \text{Tr}_{S,n+p} \left\{ b_{n+p}^{\dagger} U_{n+p} \left( \mathcal{E}_S^{t_{n+p}-t_n} \{[\rho_S(t_n), \sigma_-]\} \otimes \eta_{n+p} \right) U_{n+p}^{\dagger} \right\} \\
&+ \sqrt{\gamma \Delta t} \langle \sigma_- \rangle_{t_n} \text{Tr}_{S,n+p} \left\{ b_{n+p}^{\dagger} U_{n+p} \left( \rho_S\left(t_{n+p}\right) \otimes \eta_{n+p} \right) U_{n+p}^{\dagger} \right\}.
\end{aligned} \tag{I.16}$$

1181  Only the first order evolution terms contribute during the second collision as the second order
1182  evolution contribution to $G_{np}^{\chi}$ are of order $o(\gamma \Delta t)$. By expanding $G_{np}^{\chi}$ in the form of Eq. (18)

1183 and Eq. (32), we find that the uncorrelated evolution terms of the second collision do not
1184 contribute. Therefore, only the first order correlating part of the evolution during the second
1185 collision contributes to the correlated part of the spectrum and by taking the trace over the
1186 second collisional unit (mode $n + p$) we find

$$
\begin{aligned}
G_{np}^{\chi} =& \gamma \Delta t \left( \mathrm{Tr} \left\{ \sigma_+ \mathcal{E}_S^{t_{n+p}-t_n} \{ \sigma_- \rho_S (t_n) \} \right\} - \bar{n}_{\mathrm{th}} \mathrm{Tr} \left\{ \sigma_+ \mathcal{E}_S^{t_{n+p}-t_n} \{ [\rho_S (t_n), \sigma_-] \} \right\} \right) \\
& - \gamma \Delta t \langle \sigma_+ \rangle_{t_{n+p}} \langle \sigma_- \rangle_{t_n}.
\end{aligned}
\tag{I.17}
$$

1187 Hence, the contribution to the correlated and uncorrelated part of the spectral flows is

$$
\begin{aligned}
\dot{S}^{\otimes} (\omega_i, t_n) =& \frac{1}{\pi} \mathrm{Re} \sum_p e^{-i(\omega_i - \omega_0) t_p} G_{np}^{\otimes} \\
=& -\frac{\sqrt{\gamma \Delta t}}{\pi} \mathrm{Re} \sum_p e^{-i(\omega_i - \omega_0) t_p} \left( \langle \sigma_- \rangle_{t_n} \langle b_{n+p}^{\dagger} \rangle_{t_{n+p}} + \langle b_n \rangle_{t_n} \langle \sigma_+ \rangle_{t_{n+p}} \right) \\
& + \frac{\gamma \Delta t}{\pi} \mathrm{Re} \sum_p e^{-i(\omega_i - \omega_0) t_p} \langle \sigma_+ \rangle_{t_{n+p}} \langle \sigma_- \rangle_{t_n},
\end{aligned}
\tag{I.18}
$$

$$
\begin{aligned}
\dot{S}^{\chi} (\omega_i, t_n) =& \frac{1}{\pi} \mathrm{Re} \sum_p e^{-i(\omega_i - \omega_0) t_p} G_{np}^{\chi} \\
=& \frac{\gamma \Delta t}{\pi} \mathrm{Re} \sum_p e^{-i(\omega_i - \omega_0) t_p} \left( \mathrm{Tr} \left\{ \sigma_+ \mathcal{E}_S^{t_{n+p}-t_n} \{ \sigma_- \rho_S (t_n) \} \right\} \right. \\
& \left. \hspace{4cm} - \bar{n}_{\mathrm{th}} \mathrm{Tr} \left\{ \sigma_+ \mathcal{E}_S^{t_{n+p}-t_n} \{ [\rho_S (t_n), \sigma_-] \} \right\} \right) \\
& - \frac{\gamma \Delta t}{\pi} \mathrm{Re} \sum_p e^{-i(\omega_i - \omega_0) t_p} \langle \sigma_+ \rangle_{t_{n+p}} \langle \sigma_- \rangle_{t_n}.
\end{aligned}
\tag{I.19}
$$

1188 We rewrite the expressions using only atomic operators, using $\langle b_{\mathrm{in}} (t) \rangle = \frac{\Omega}{2\sqrt{\gamma}} e^{-i(\omega_L - \omega_0)t}$ and
1189 $\langle \sigma_+ \rangle_{t_{n+p}} = e^{i(\omega_L - \omega_0) t_p} \langle \sigma_+ \rangle_{t_n}$ as the atom is in its limit cycle. Taking the continuous limit of $\dot{S}^{\otimes}$
1190 and $\dot{S}^{\chi}$, this leads to the following expressions:

$$
\begin{aligned}
\dot{S}^{\otimes} (\omega, t_\infty) =& -\delta_D (\omega - \omega_L) \frac{\Omega}{2} \left( e^{i(\omega_L - \omega_0)t} \langle \sigma_- \rangle_{t_\infty} + e^{-i(\omega_L - \omega_0)t} \langle \sigma_+ \rangle_{t_\infty} \right) \\
& + \delta_D (\omega - \omega_L) \gamma \left| \langle \sigma_- \rangle_{t_\infty} \right|^2
\end{aligned}
\tag{I.20}
$$

$$
\begin{aligned}
\dot{S}^{\chi} (\omega, t_\infty) =& \frac{\gamma}{\pi} \mathrm{Re} \int_0^{t_\infty} d\tau e^{-i(\omega - \omega_0)\tau} \left( \mathrm{Tr} \left\{ \sigma_+ \mathcal{E}_S^{\tau} \{ \sigma_- \rho_S (t_\infty) \} \right\} \right. \\
& \left. \hspace{4cm} - \bar{n}_{\mathrm{th}} \mathrm{Tr} \left\{ \sigma_+ \mathcal{E}_S^{\tau} \{ [\rho_S (t_\infty), \sigma_-] \} \right\} \right) \\
& - \frac{\gamma}{\pi} \mathrm{Re} \int_0^{t_\infty} d\tau e^{-i(\omega - \omega_0)\tau} \langle \sigma_+ \rangle_{t_\infty + \tau} \langle \sigma_- \rangle_{t_\infty}.
\end{aligned}
\tag{I.21}
$$

1191 Hence, we find Eq. (61) by setting $\omega_L = \omega_0$ in $\dot{S}^{\otimes} (\omega) \equiv \dot{S}^{\otimes} (\omega, t_\infty)$. The inelastic component
1192 of the Mollow triplet can be recovered by setting $\omega_L = \omega_0$ and $\bar{n}_{\mathrm{th}} = 0$ in $\dot{S}^{\chi}(\omega) \equiv \dot{S}^{\chi}(\omega, t_\infty)$.

Similarly the input photon spectral rate takes the form:

$$
\begin{aligned}
\dot{S}_{\mathrm{in}}(\omega, t) =& \frac{1}{\pi} \mathrm{Re} \int_0^{t_\infty} d\tau \, e^{-i\omega\tau} e^{i\omega_0\tau} \left\langle b_{\mathrm{in}}^\dagger(t+\tau) \right\rangle_{t+\tau} \left\langle b_{\mathrm{in}}(t) \right\rangle_t \\
&+ \frac{1}{\pi} \mathrm{Re} \int_0^{t_\infty} d\tau \, e^{-i(\omega-\omega_0)\tau} \delta_D(\tau) \left\langle b_{\mathrm{in}}^\dagger(t+\tau) b_{\mathrm{in}}(t) \right\rangle_0 \\
=& \, \delta_D(\omega_L - \omega) \frac{\Omega^2}{4\gamma} + \frac{1}{2\pi} \bar{n}_{\mathrm{th}}.
\end{aligned}
\tag{I.22}
$$

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
