# Peer review of "Thermodynamics of autonomous optical Bloch equations"

_SciPost Physics_

## Round 1 · Referee Report · Anonymous (Referee 1) · 2025-12-21

Strengths

1) The paper deals with a fundamental issue of a the Thermodynamics of optical Bloch equations. The Bloch equation is the template of the quantum Master equation. Originally guessed by Bloch it has since been derived from first principles. 2) The paper addresses specifically an autonomous derivation meaning the the drive is also a quantum system. 3) The authors show the connection to a collision model one of the rigorous approaches to derive the Master equation. 4) Definitions of heat and work are obtained connecting to thermodynamics. 5) the derivation is clear.

Weaknesses

1) The authors overlook closely related work on an autonomous driven master equation:

Dann, et. al. "Quantum thermo-dynamical construction for driven open quantum systems." Quantum 5 (2021): 590.

Dann, et al. "Unification of the first law of quantum thermodynamics." New Journal of Physics 25, no. 4 (2023): 043019

Report

The authors should specify the relation to other autonomous constructions
of a driven Master equation.

Recommendation

Ask for major revision

---

## Editorial Decision

in_refereeing